# Weaker neural suppression in autism

Michael-Paul Schallmo [1,2✉], Tamar Kolodny [1], Alexander M. Kale [1], Rachel Millin[1], Anastasia V. Flevaris[1], Richard A. E. Edden[3], Jennifer Gerdts[4], Raphael A. Bernier[4] & Scott O. Murray[1]

Abnormal sensory processing has been observed in autism, including superior visual motion discrimination, but the neural basis for these sensory changes remains unknown. Leveraging well-characterized suppressive neural circuits in the visual system, we used behavioral and fMRI tasks to demonstrate a significant reduction in neural suppression in young adults with autism spectrum disorder (ASD) compared to neurotypical controls. MR spectroscopy measurements revealed no group differences in neurotransmitter signals. We show how a computational model that incorporates divisive normalization, as well as narrower top-down gain (that could result, for example, from a narrower window of attention), can explain our observations and divergent previous findings. Thus, weaker neural suppression is reflected in visual task performance and fMRI measures in ASD, and may be attributable to differences in top-down processing.

[1] Department of Psychology, University of Washington, UW Box 351525, Seattle, WA 98195, USA. [2] Department of Psychiatry and Behavioral Sciences, University of Minnesota, 2450 Riverside Avenue S, Minneapolis, MN 55454, USA. [3] Department of Radiology and Radiological Science, Johns Hopkins University, 600 N Wolfe St, Baltimore, MD 21287, USA. [4] Department of Psychiatry and Behavioral Sciences, University of Washington, UW Box 356560, Seattle, WA 98195, USA. ✉email: schal110@umn.edu

Autism spectrum disorder (ASD) is a neurodevelopmental condition characterized by social difficulties, repetitive behaviors, and sensory abnormalities[1], the cause of which remains unknown. Recent efforts to identify underlying neuro-computational changes in ASD have targeted sensory and perceptual systems[2–9] as they feature prominently in symptomatology[1], are amenable to comprehensive psychophysical measurement, can be directly tied to electrophysiological findings in animals, and can be modeled using well-established principles of cortical computation. For example, large enhancements in visual motion discrimination performance found in ASD compared to neurotypical (NT) controls[4] were recently described as a deficit in normalization[2]—a computation that reflects neural processes which regulate (i.e., suppress) neural responses in the brain[10,11]. Under this hypothesis, weaker normalization in ASD would result in larger amplitude neural responses and lead to enhanced behavioral performance in tasks that depend on neural sensitivity, such as motion discrimination. Although the neural basis of enhanced motion perception in ASD is not yet clear, the neural mechanisms of motion discrimination are well studied in both humans[12,13] and animal models[14,15], which may help to pinpoint neural differences in ASD.

Unfortunately, inconsistent experimental findings in the ASD sensory-perceptual literature have made it difficult to identify underlying neuro-computational changes to date. For example, individuals with ASD do not perform uniformly better on perceptual tasks that depend on neural sensitivity (for a review, see ref. [16]), as predicted by the weaker normalization hypothesis. In fact, two recent studies have demonstrated higher motion discrimination thresholds (worse performance) for high-contrast gratings in ASD compared to controls[3,9]—a reversal of the originally observed effect[4]. Experimental variability in ASD is often explained in terms of the heterogeneity of the autism spectrum. However, to be considered viable, a computational account of abnormal sensory processing in autism should be able to encompass such variability and should be supported by neurophysiological evidence.

In the current study, we sought to determine a neural basis for abnormal motion perception in ASD and to find a computational account that satisfactorily describes our own behavioral results as well as those of previous studies[3,4,9]. We examined visual motion processing in ASD at both a behavioral and neural level, using visual psychophysics and functional magnetic resonance imaging (fMRI) respectively. In addition, we used MR spectroscopy (MRS) to measure neurotransmitter levels in vivo, in order to probe the role of inhibition during visual perception in ASD. Our results indicate that ASD is associated with abnormally weak neural suppression within the motion-sensitive brain area called human middle temporal complex[17,18] (hMT+), but we found no difference in GABA levels in this region. Previous computational models failed to account for our findings of weaker spatial suppression in ASD. Instead, we show that a model which incorporates divisive normalization and narrower top–down gain provides a parsimonious computational basis for the observed reduction in suppression in ASD. Finally, using this model we show how variability in the width of top–down gain across individuals with ASD (while always remaining smaller than NTs), as well as interactions with stimulus size and contrast, can potentially account for discrepant findings of both enhanced and impaired sensory processing in this disorder.

## Results

**Foreword**. To assess suppressive modulatory mechanisms in humans with ASD, we measured visual spatial suppression, a phenomenon in which larger moving stimuli are more difficult to perceive[19]. This mirrors a well-known neural phenomenon; when stimuli extend beyond a neuron's spatial receptive field, neural responses in visual cortex are suppressed through a combination of feed-forward, lateral, and feedback interactions[20–22]. Based on work in both humans[12,23,24] and non-human primates[14,25–27], it is thought that neural surround suppression within the motion-selective visual area MT plays an important role in the perceptual phenomenon of spatial suppression during motion discrimination. In a series of 3 experiments within the current study, we characterized spatial suppression using behavioral, neural, and neurochemical methods in a group of 28 young adults with ASD and a comparison group of 35 NT participants (for demographic information, see Table 1). We then present a computational account of weaker spatial suppression in ASD in the context of a divisive normalization model.

**Behavior**. We obtained a quantitative behavioral index of spatial suppression by measuring motion duration thresholds[19]. It is known that the amount of time that a stimulus needs to be presented in order to perceive motion direction depends on stimulus size; paradoxically, larger stimuli require longer presentation durations[13]. In our task, participants judged whether visual grating stimuli drifted left or right (Fig. 1a–c). Motion duration thresholds were defined by the minimum stimulus duration for which participants could perceive motion direction with 80% accuracy. Thresholds were measured for each of the three different stimulus sizes and two different contrasts (Fig. 1a, b). Note that this task does not depend on reaction time; although stimulus duration was brief (Fig. 1c), response time was not limited.

We observed the expected spatial suppression effect; thresholds were significantly longer for larger stimuli (main effect of size; $F_{1,61} = 44.1$, $p = 9 \times 10^{-9}$; Fig. 2a, b). Importantly, spatial suppression was significantly weaker in the ASD group vs. NTs (group × size interaction; $F_{1,61} = 9.76$, $p = 0.003$). Thus, as stimulus size increased, duration thresholds increased less dramatically among participants with ASD. Weaker spatial

**Table 1 Participant demographics.**

| Demographics | ASD (n = 28) | NT (n = 35) | Statistics |
|---|---|---|---|
| Age in years | 22.4 (3.44) | 23.4 (3.56) | $t_{(61)} = 1.09$, $p = 0.3$ |
| Biological sex | 18 M; 10 F | 21 M; 14 F | $X^2_{(1)} = 0.121$, $p = 0.7$ |
| Non-verbal IQ[a] | 112 (17.6) | 113 (13.2) | $t_{(61)} = 0.418$, $p = 0.7$ |
| Handedness | 4 L; 24 R | 2 L; 33 R | $X^2_{(1)} = 0.518$, $p = 0.5$ |
| SRS-2 score[b] | 72.1 (27.0) | 38.7 (17.8) | $t_{(61)} = 5.91$, $p = 2 \times 10^{-7}$ |
| ADOS-2 score[c] | 7.32 (1.59) | — | |

Values shown are group mean and SD. Results of statistical tests for group differences are shown on the right. Two-sample t tests and chi-square tests were used to assess group differences.
[a]Non-verbal IQ was calculated based on the Wechsler Abbreviated Scale of Intelligence (WASI).
[b]Social Responsiveness Scale, 2nd edn. (SRS-2) total score.
[c]Autism Diagnostic Observation Schedule, 2nd edn. (ADOS-2) comparison score.

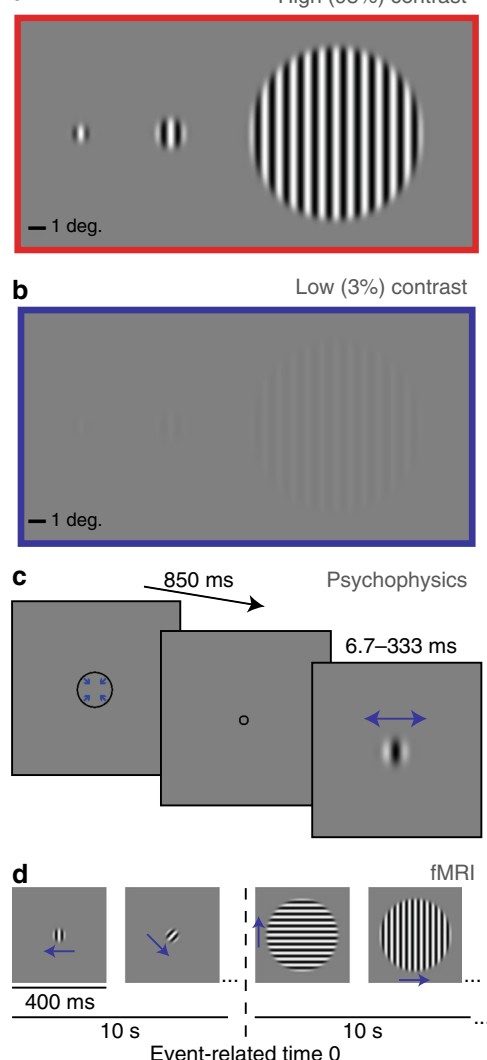

**Fig. 1 Stimuli and paradigms. a** Small (0.84°), medium (1.7°), and big gratings (10° diameter) at high contrast (98%). **b** The same gratings at low contrast (3%). **c** Psychophysical paradigm: a fixation mark (shrinking circle) was followed by a briefly presented drifting grating (left or right; small high-contrast grating shown). Blue arrows indicate direction of motion. **d** fMRI paradigm: alternating 10 s blocks of smaller (2°) and larger (12°) gratings (high contrast shown). Vertical dashed line indicates the transition from smaller to larger, which is the event of experimental interest. Note that stimuli are scaled differently across panels for display purposes.

suppression in ASD did not depend on stimulus contrast; other interactions and main effects, including a main effect of group, were also not significant (all $F_{1,61} < 1.78$, $p$ values > 0.19).

To quantify spatial suppression we computed size indices, which involved taking the logarithm and then the difference between thresholds for medium and large stimuli (see Eq. (3) in Methods). More negative size indices reflect stronger suppression (greater increase in duration thresholds with increasing stimulus size; Fig. 2c). Comparing size indices between groups showed the same effect of weaker spatial suppression in participants with ASD vs. NTs (main effect of group; $F_{1,61} = 9.37$, $p = 0.003$). In this case, we found size indices in both groups were more negative for low- vs. high-contrast gratings, indicating stronger suppression (main effect of contrast; $F_{1,61} = 19.7$, $p = 4 \times 10^{-5}$). Weaker suppression in ASD did not depend on stimulus contrast (group × contrast interaction; $F_{1,61} = 1.55$, $p = 0.2$). Together,

these results indicate that our participants with ASD experienced weaker spatial suppression; they were able to perceive the direction of motion for stimuli presented more briefly, especially when those stimuli were large (the most challenging condition). Weaker spatial suppression in our behavioral task suggests that neural suppression in visual cortex may also be weaker in ASD.

**Functional MRI**. We used an fMRI paradigm designed to measure spatial suppression in order to probe neural suppression in ASD more directly. In NT participants, we have recently shown that this paradigm, which involves presenting alternating blocks of smaller and larger drifting gratings (Fig. 1d), yields suppressed fMRI responses for larger vs. smaller stimuli within foveal regions of visual cortex[12]. Participants performed a colored shape detection task at fixation, in order to minimize eye movements, divert attention away from the drifting gratings, and emphasize bottom–up stimulus processing.

We first examined the fMRI response within the motion-selective region of the lateral occipital lobe known as human MT complex (hMT+; Supplementary Figs. 1 and 2). Because hMT+ is retinotopically organized[28], and because our stimuli were presented at the fovea, we focused on voxels that showed significant selectivity for foveal over peripheral stimuli (see Methods; Supplementary Fig. 1). For both low- and high-contrast stimuli, fMRI responses in foveal hMT+ to larger stimuli were significantly below baseline (i.e., lower than responses to the preceding smaller stimuli; paired $t$ tests, $t_{47} > 4.13$, $p$ values < 0.003, Bonferroni corrected for 2 comparisons; Fig. 2d, e). Thus we saw robust suppression of the fMRI signal in hMT+ in response to increasing stimulus size, in agreement with the spatial suppression effect observed in our motion discrimination task above. Importantly, fMRI suppression within hMT+ was significantly weaker for ASD vs. NT participants (main effect of group, $F_{1,47} = 5.66$, $p = 0.022$; Fig. 2f). Suppression in both groups was stronger for high- vs. low-contrast stimuli (main effect of contrast, $F_{1,47} = 5.52$, $p = 0.023$), but weaker suppression in ASD did not depend on contrast (group × contrast interaction, $F_{1,47} = 1.15$, $p = 0.3$). These findings indicate that neural suppression in foveal hMT+ is weaker for participants with ASD compared to their NT peers, in agreement with our behavioral results.

We did not observe a correlation between fMRI suppression in hMT+ and psychophysical suppression indices across individuals ($r_{47} = 0.02$, $p = 0.9$ for all participants). This may be attributed to differences in attention (i.e., gratings were attended during psychophysics, while attention was directed toward the fixation task during fMRI), the involvement of additional brain areas beyond hMT+ during motion perception (e.g., V1, higher-level regions), the fact that fMRI and psychophysical data were collected in separate experimental sessions, and/or slight stimulus differences (e.g., above-threshold stimulus duration during fMRI).

Next, we examined fMRI responses within a region of early visual cortex (EVC; at the foveal confluence of V1, V2, and V3 near the occipital pole). We have previously found that, among NT individuals, fMRI responses in foveal EVC are also suppressed by larger vs. smaller stimuli but that the pattern of fMRI suppression in hMT+ was a better match to the spatial suppression observed psychophysically[12]. Here again, fMRI responses in EVC were significantly suppressed below baseline for both high- and low-contrast stimuli (paired $t$ tests, $t_{54} > 9.04$, $p$ values < $4 \times 10^{-11}$, Bonferroni corrected for 2 comparisons; Fig. 2g, h), in agreement with the expected spatial suppression effect. However, unlike in hMT+, we found no significant difference in fMRI suppression within EVC between participants with ASD and NTs (main effect of group, $F_{1,55} = 1.61$, $p = 0.2$; Fig. 2i). Suppression in EVC was

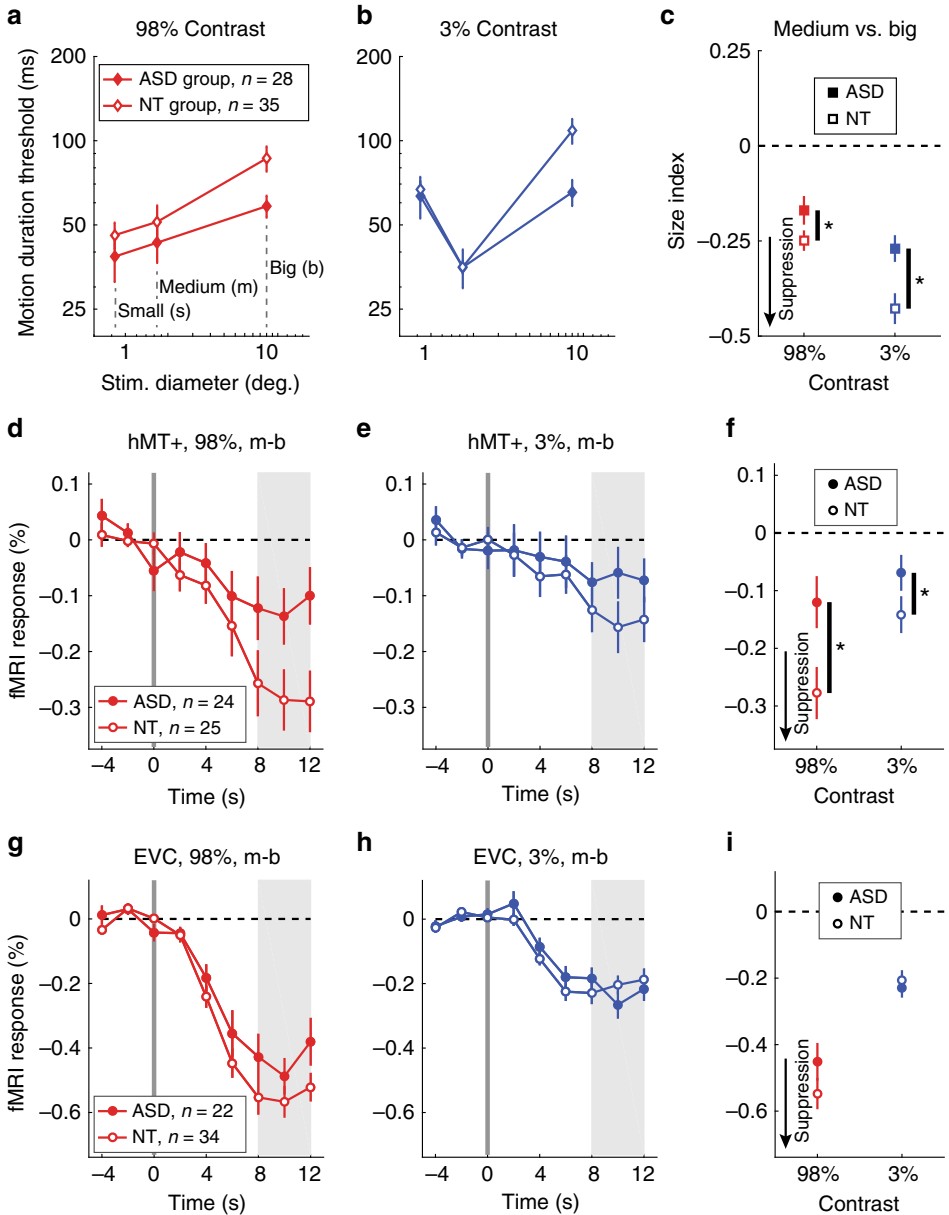

**Fig. 2 Behavioral and fMRI results. a** Motion duration thresholds for high-contrast (98%) drifting gratings. The time required to perceive whether stimuli drifted left or right with 80% accuracy is shown on the y-axis for different stimulus sizes (x-axis). **b** The same, but for low-contrast gratings (3%). **c** The effect of increasing stimulus size on duration thresholds was quantified using a size index—the difference between log thresholds (Eq. 3). More negative values indicate stronger suppression (black arrow). Participants with ASD show weaker spatial suppression during motion discrimination than NTs (asterisk: ANOVA, main effect of group, $p = 0.003$). **d** fMRI responses in foveal human MT complex (hMT+) to an increase in stimulus size; at time = 0 s, high-contrast drifting gratings increased in size from medium (m) to big (b). **e** The same, but for low-contrast gratings. **f** Average fMRI responses for each group (from shaded regions in **e**, **f**). Suppression of the fMRI response in hMT+ is weaker among participants with ASD (asterisk: ANOVA, main effect of group, $p = 0.022$). **g**–**i** are the same as **d**–**f** but for a foveal region of early visual cortex (EVC). Samples sizes for each row are shown in **a**, **d**, and **g**. Dots show group means; error bars are S.E.M.

stronger for high- vs. low-contrast stimuli (main effect of contrast, $F_{1,55} = 56.5$, $p = 7 \times 10^{-10}$), but there was no significant interaction between group and contrast (group × contrast interaction, $F_{1,53} = 1.50$, $p = 0.2$). These results indicate that fMRI responses within EVC reflect spatial suppression, but unlike for motion discrimination or responses in hMT+, there was no difference in fMRI suppression within EVC between participants with ASD and NTs.

**MR spectroscopy**. Next, we sought to determine whether weaker suppression in ASD might be attributable to differences in

excitatory or inhibitory neural functioning (e.g., weaker inhibition), as it has been suggested that there may be an imbalance of excitation and inhibition in this disorder[29–31]. We used MRS to measure the concentration of GABA+ (GABA, an inhibitory neurotransmitter, plus co-edited macromolecules) in a region centered around hMT+ (Supplementary Fig. 2a, b). However, we found no significant group difference in GABA+ within the hMT+ region nor any correlations with behavioral or fMRI suppression metrics (Supplementary Fig. 2c–h). Additional measurements of GABA+ in EVC (Supplementary Fig. 3), as well as Glx (glutamate, an excitatory neurotransmitter, plus glutamine and glutathione) in both hMT+ and EVC (Supplementary Figs. 2

and 3) also revealed no differences between participants with ASD and NTs, and no significant correlations were found with measures of spatial suppression. Given this lack of significant findings, we are not able to make any strong conclusions regarding the role of either GABA+ or Glx within visual cortex during spatial suppression among people with ASD.

**Relation to clinical measures**. To probe whether weaker spatial suppression in ASD is related to clinical functioning, we first examined correlations between the total comparison score on the Autism Diagnostic Observation Schedule, Second Edition (ADOS-2)[32] and behavioral or fMRI measures of suppression. No significant correlations between ADOS-2 total comparison scores and either size indices or fMRI suppression in hMT+ were observed in our participants with ASD ($|r_{22-26}| < 0.18$, uncorrected $p$ values > 0.3).

Abnormal sensory experiences are common among people with ASD[1]. Therefore, we also examined whether aspects of sensory processing in everyday life, as measured by the Sensory Profile[33,34] subscales for sensitivity and avoiding (summed scores), were associated with suppression metrics in both ASD and NT participants. These measures reflect ranked self-reported responses to questions such as "I am bothered by unsteady or fast-moving images." We observed a moderate correlation between higher sensory sensitivity + avoiding and weaker fMRI suppression in hMT+ ($r_{46} = 0.34$, uncorrected $p = 0.019$, Bonferroni corrected for 4 multiple comparisons between suppression metrics and symptom scores $p = 0.074$; Fig. 3). However, no significant relationship with behavioral size indices was found ($r_{60} = -0.004$, uncorrected $p = 0.98$). The former result may suggest that weaker neural suppression within hMT+ could be relevant to sensory dysfunction during daily life.

**Computational modeling**. We next examined different computational principles that might account for weaker neural suppression in ASD. Recent work[2,12] has suggested that a general computational model for spatial vision, known as divisive normalization (Fig. 4), can describe the effect of stimulus size on motion duration thresholds measured psychophysically. Divisive normalization models have been used to describe the effects of spatial context on neural responses in visual cortex, which can be summarized by saying: a neuron's response is divided by the summed response of its neighbors[10,35]. Rosenberg and colleagues[2] have proposed that a reduction in divisive normalization might account for abnormal performance across a number of visual tasks in people with ASD. Weaker normalization in ASD yields a model predicting overall superior motion discrimination performance (i.e., lower duration thresholds), consistent with some previous behavioral findings[4] but not others[3,9]. An alternative computational model has been presented by Schauder and colleagues[3], who suggested that larger excitatory spatial filters (SFs) could account for their observation of higher motion duration thresholds overall among young people with ASD vs. NTs (see also ref. [9]). We have recently used a divisive normalization model to describe spatial suppression across a series of experiments in NT participants[12]. Here we expand upon our modeling work by considering three different modifications, including variants based on the two models noted above[2,3] (for full modeling details, see Methods and Supplementary Methods). By comparing different versions of the divisive normalization model, we sought to: (1) describe our current observations of weaker spatial suppression in ASD within a normalization model framework and (2) find a model in which a consistent parameter difference between groups is capable of predicting both superior and reduced motion duration thresholds in ASD, given that

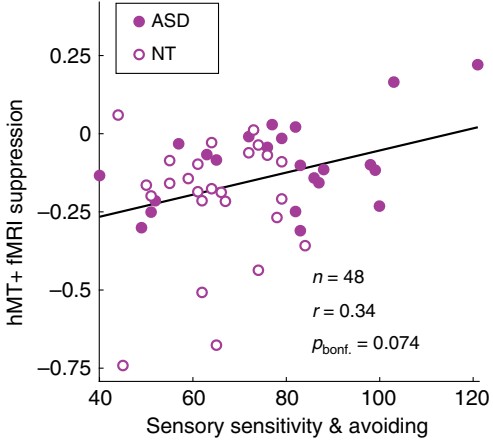

**Fig. 3 Relationship with sensory dysfunction.** Weaker fMRI suppression in foveal hMT+ was associated with higher combined scores for sensory sensitivity and avoiding from the Sensory Profile, across both ASD and NT participants. Black line indicates linear trend. Correlation is Pearson's $r$ value.

behavioral results across previous studies have disagreed[3,4,9], as noted above.

We first asked whether weaker normalization strength (i.e., a 25% reduction in suppressive gain[2]; Fig. 4, orange box; Supplementary Table 2) could describe the difference in spatial suppression we observed psychophysically between individuals with ASD and NTs. We found that weaker normalization reduced the motion duration thresholds predicted by the model (Fig. 5a, b). Critically however, this reduction in motion thresholds was not specific to the largest stimuli but was observed across all stimulus sizes. Thus weaker normalization had little effect on spatial suppression (red arrows, Fig. 5c). In this way, the model proposed by Rosenberg and colleagues[2] failed to account for our behavioral and fMRI findings of weaker spatial suppression in ASD (Fig. 2a–c; see also Supplementary Fig. 5). Therefore, we conclude that a reduction in suppressive gain within the normalization model[2] is not sufficient to account for our observations of weaker spatial suppression in participants with ASD.

Next, we examined whether larger excitatory SFs (25% wider[3]; Fig. 4, magenta box; Supplementary Table 2) could account for weaker spatial suppression within our model framework. Indeed, we found that larger excitatory SFs did predict weaker spatial suppression (Fig. 5f). However, this was driven by larger predicted duration thresholds for small- and medium-sized stimuli (red arrows, Fig. 5d, e; Supplementary Fig. 5), unlike the pattern of results we observed in participants with ASD (smaller thresholds for large stimuli; Fig. 2a, b). Therefore, we find that larger excitatory SFs[3] within the framework of our normalization model are not sufficient to explain the pattern of motion duration threshold data we observed in people with ASD.

Finally, we considered whether differences in top–down gain modulation (that could, for example, reflect differences in spatial attention), as described by the normalization model, might better account for our observations of weaker spatial suppression in ASD (Fig. 2a–c). It has been suggested that the focus of spatial attention may be narrower in people with ASD[5], and top–down effects (such as attention or expectation) that modulate the gain of neural processing can be modeled in terms of divisive normalization[35].

We found that a narrower top–down gain field within our model (6 vs. 14 arbitrary units; Fig. 4, cyan box; Supplementary

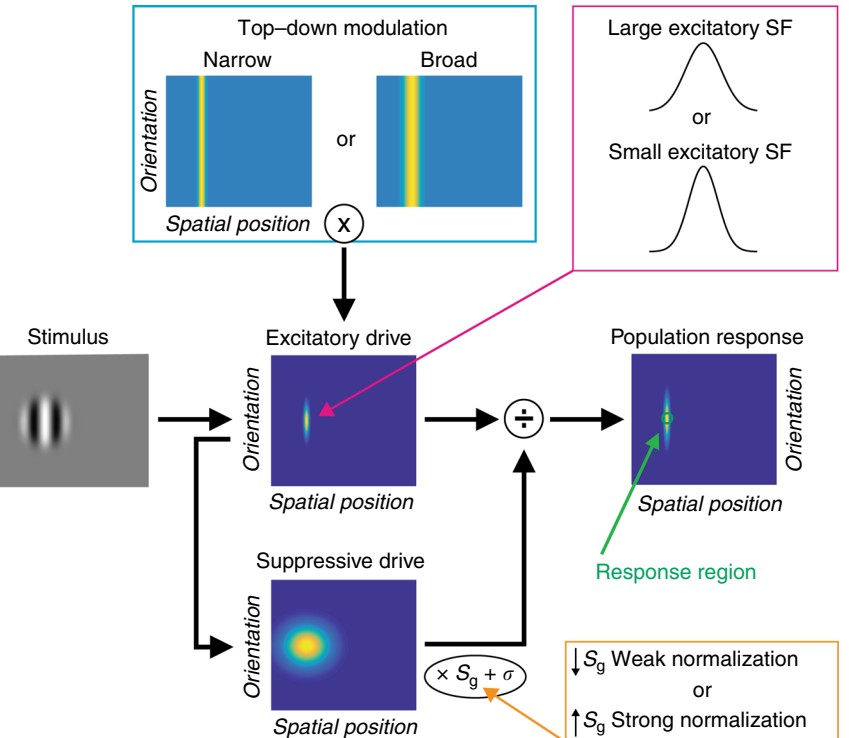

**Fig. 4 Normalization model diagram.** For full model details, see Methods and Supplementary Methods. We tested three different model variants, which are described within colored boxes. Orange: Weaker vs. stronger normalization was modeled by lower vs. higher values for the suppressive gain term ($S_g$), as in previous work[2]. Magenta: Large vs. small excitatory spatial filters (SFs) were modeled using wider vs. narrower spatial Gaussians in the excitatory drive term ($E$), as in previous work[3]. Cyan: Narrow vs. broad top–down modulation was modeled using narrower vs. broader spatial Gaussians in the top–down modulation term ($M$).

Table 2) led to predicted motion duration thresholds that were smaller, especially for larger stimuli (Fig. 5g, h). Model size indices were likewise less negative with narrower top–down modulation (Fig. 5i), indicating weaker spatial suppression. Narrower top–down gain modulation within the normalization model therefore predicts a pattern of results that closely mirrored our observation of weaker spatial suppression in ASD (compare Fig. 5g–i with Fig. 2a–c). In this way, our narrower top–down gain model provided a qualitatively better match to our behavioral results (showing weaker spatial suppression in ASD), as compared to the predictions of previously published models[2,3] (see also Supplementary Fig. 5).

Unlike low-level neural properties such as normalization strength or excitatory SF size, top–down neural processes such as spatial attention may be expected to vary greatly as a function of the task being performed. Thus we asked whether variability in top–down modulation width might be sufficient to account for the inconsistent results of both superior and reduced motion perception in ASD. Indeed, we found that using even narrower top–down parameters (1 or 2 arbitrary units for ASD vs. 6 for NTs) yielded model predictions that showed similarities to previous observations of both lower[4] or higher[3,9] motion duration thresholds in people with ASD (depending on stimulus size and contrast; Supplementary Fig. 4). Specifically, our model predicts higher thresholds when the width of top–down modulation is narrower than the neural population that is sampled for the perceptual decision (i.e., in Fig. 4, when the top–down modulation [cyan box] is narrower than the response region [green arrow]).

In comparison, models from previous studies[2,3] have sought to predict either lower or higher motion duration thresholds (but not both). To account for both superior[4] and reduced[3,9] motion

discrimination, previous models would suggest that neural differences between participants with ASD and NTs are not consistent across experimental samples. Normalization, for example, would need to be weaker in ASD to explain lower motion duration thresholds in one sample[2,4], but stronger in ASD in another sample to account for higher thresholds[3,9]. Our model may be more parsimonious, as the width of top–down gain is always modeled as narrower in ASD vs. NT participants; whether higher or lower thresholds are predicted in ASD depends on the width of top–down modulation relative to the size of the neural population that is used for the perceptual decision (Fig. 4; Supplementary Fig. 4). In summary, a model that incorporates narrower spatial top–down gain modulation within the divisive normalization framework may provide a unified computational account for our observations of weaker spatial suppression among participants with ASD vs. NTs, as well as previous divergent findings[2–4,9].

**Control analyses.** Finally, we performed a series of control analyses to rule out alternative explanations for our results showing weaker behavioral and fMRI suppression in the ASD group (Fig. 2a–f). Although demographic factors did not differ significantly between groups, previous work has shown that age[36–40], biological sex[41], and intelligence quotient (IQ)[41–43] may each be associated with differences in motion duration thresholds and/or the magnitude of spatial suppression. We sought to control for these factors in post hoc analyses by including them as covariates when testing for group differences in behavioral or fMRI suppression. Our results were unaffected by including age, sex, and IQ as factors in linear mixed-effects models; suppression was still significantly weaker among participants with ASD vs. NTs in

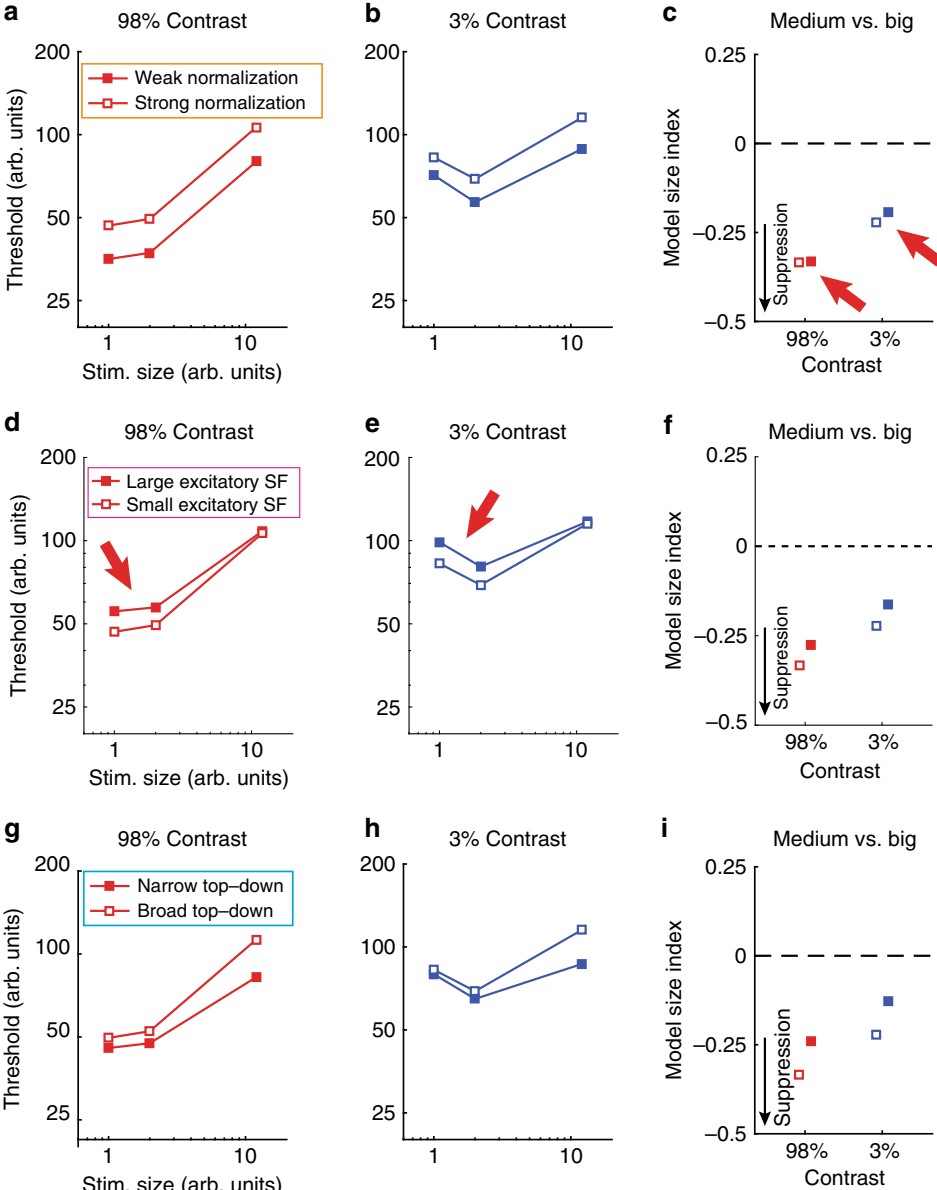

**Fig. 5 Normalization model results.** We used a divisive normalization model to describe motion duration thresholds, as in our previous work[12].
**a**, **b** Weaker normalization (25% weaker suppressive gain[2]) yields lower predicted duration thresholds across stimulus sizes and contrasts. **c** Size indices are not dramatically altered by weaker normalization. More negative values indicate stronger suppression (black arrow). Red arrows indicate a poor match to our behavioral results from Fig. 2a–c. **d–f** Larger excitatory spatial filters (25% larger[3]) yield higher duration thresholds and less negative size indices. Thus neither weak normalization nor larger excitatory spatial filters provide a good match for our observations of lower motion discrimination thresholds for large stimuli among people with ASD (Fig. 2a–c). **g**, **h** Narrower top–down modulation (6 vs. 14 arb. units) yields lower thresholds at larger stimulus sizes. **i** Narrower top–down modulation produces less negative size indices, consistent with weaker suppression, and thus shows a better match to our psychophysical results.

both the motion discrimination task and in the fMRI response within hMT+ (see Supplementary Notes).

We also asked whether our fMRI results might be explained by differences in head motion or fixation task performance between groups. Head motion during fMRI (mean framewise displacement[44]) was significantly greater among participants with ASD vs. NTs (ASD mean = 0.15 mm, SD = 0.09, NT mean = 0.09 mm, SD = 0.05; Mann–Whitney test, Z = 2.64, p = 0.008), but we saw no significant group difference in hit rates during the fixation task (ASD mean = 95.0%, SD = 7.0%, NT mean = 93.9%, SD = 8.1%; analysis of variance (ANOVA), main effect of group, $F_{1,61} = 0.31$, $p = 0.6$). To further address this question, we excluded fMRI data with excessive head motion or poor fixation task performance

(see Supplementary Notes). However, the results were qualitatively the same as those shown in Fig. 2g–i; after exclusion, fMRI response suppression within hMT+ was weaker among participants with ASD vs. NTs (Supplementary Fig. 6).

Further, we examined whether differences in eye movements between groups might explain weaker spatial suppression in participants with ASD. We found no significant differences in eye movement metrics between groups and no correlations between these metrics and our measures of suppression from psychophysics or from fMRI (see Supplementary Notes and Supplementary Methods).

Last, we explored whether MRS data quality may have varied systematically between groups. For example, if data were

generally poorer among participants with ASD, then this might possibly have obscured the pattern of results in our MRS experiments. However, we found that data quality metrics (e.g., water frequency variability, number of repetition times (TRs) rejected for outliers) were generally comparable between NT and ASD groups (see Supplementary Notes), suggesting that our MRS results were not confounded by data quality differences. Overall, the results of our control analyses do not suggest that demographic or data quality differences may account for the patterns of results we have observed in ASD vs. NT participants.

## Discussion

We have found evidence for weaker neural suppression in people with autism. Specifically, spatial suppression, a phenomenon we measured in both a visual motion discrimination task and using fMRI in the visual area hMT+, was significantly weaker among participants with ASD compared with demographically matched NT individuals (Fig. 2a–f). These behavioral results lend support to previous studies; our observations of lower motion duration thresholds (superior discrimination, for large stimuli) generally agree with the findings of Foss-Feig and colleagues[4], while weaker spatial suppression is generally consistent with the observations of Sysoeva and colleagues[9]. Importantly, our fMRI data recapitulated the pattern of weaker spatial suppression in ASD that we observed behaviorally. These results provide insight into the neurophysiological basis of this phenomenon and indicate that motion perception anomalies in ASD may depend, at least in part, on weaker spatial suppression within hMT+, a visual motion-selective region in the lateral occipital lobe.

Our computational modeling work suggests that a possible higher-level mechanism may underlie weaker suppression in ASD: a top–down process (such as spatial attention) that amplifies neural responses in visual cortex (e.g., within area hMT+) may be tuned more narrowly in space among persons with ASD (Fig. 4, cyan box; Fig. 5g–i). The reason that narrower top–down gain can result in weaker neural suppression may be understood intuitively by referring to Fig. 4 (cyan box). When stimuli are small, whether the top–down gain is narrow or broad has little effect on the model behavior; the suppressive drive and the excitatory drive it depends on are both similarly engaged. However, when a stimulus is large, broader top–down gain results in stronger engagement of the suppressive drive (which is spatially broad), yielding more drastic increases in predicted thresholds for larger stimuli (i.e., stronger suppression). Likewise, narrower top–down gain predicts weaker suppression.

Narrower top–down neural gain could, for example, reflect intrinsic differences in spatial attention—individuals with autism may have narrower windows of attention compared to NT individuals. While this is consistent with previous experimental findings showing a sharper gradient of spatial attention in ASD[5] and is consistent with detail-focused perceptual behavior associated with autism[16,45,46], we did not explicitly manipulate spatial attention in the current study. Thus we can only speculate about the cognitive origins of the narrower spatial gain suggested by the model. Importantly, the model does not imply any impairment or reduction in spatial attention in ASD per se, only a small difference in how attention is allocated in space. This difference in allocation could reflect an inclination toward local processing, as is sometimes invoked to characterize ASD[46] or may reflect intrinsically altered structure of feedback circuits[20,47].

The computational model we have proposed showed some advantages when compared to other models that have been used previously to describe differences in motion discrimination between participants with ASD vs. NT individuals. Rosenberg and colleagues[2] showed that weaker normalization (i.e., lower

suppressive gain term, $S_g$ in our model (Eq. (1)); Fig. 4, orange box) was sufficient to explain the overall superior motion discrimination performance (lower duration thresholds) among participants with ASD reported by Foss-Feig and colleagues[4]. Although founded on the same computational principles (i.e., divisive normalization[10,35]) as the model we have proposed, we found that the Rosenberg[2] model was not able to describe the pattern of weaker spatial suppression we observed in the current study; weaker normalization had little effect on spatial suppression (compare Fig. 2a–c with Fig. 5a–c [red arrows]; see also Supplementary Fig. 5b–d). This led us to adopt a different computational strategy (i.e., implementing narrower top–down gain modulation, rather than weaker normalization) to account for the current behavioral results.

In contrast, Schauder and colleagues[3] found equivalent or even higher duration thresholds for ASD participants, across different experimental conditions (for a similar report of higher thresholds in ASD, see ref. [9]). They used a computational model to show that larger excitatory SFs ($x_{w\_e}$ in our model; Supplementary Eq. 2; Fig. 4, magenta box) could explain higher duration thresholds in ASD. We found that this model was also not sufficient to explain our observation of weaker spatial suppression in ASD, as it predicted higher thresholds for smaller stimuli in ASD, rather than lower thresholds for larger stimuli (red arrows, Fig. 5d–f and Supplementary Fig. 5e–g). We further showed that a model based on narrower top–down gain modulation was sufficient to describe not only our findings of weaker spatial suppression in ASD (Fig. 5g–i) but may also be extended to cover disparate patterns of motion discrimination (i.e., both lower and higher thresholds) found in ASD in previous studies[2–4,9] (Supplementary Fig. 4a–f). Another aspect that distinguishes our model from previous work is that it suggests a difference in higher-level neural processes in ASD (top–down modulation of sensory responses, which might be attributed to narrower spatial attention), whereas other models have suggested lower-level differences in sensory processing (i.e., weaker normalization or wider excitatory SFs).

Weaker neural suppression in ASD was not apparent within earlier regions of visual cortex; we found weaker suppression among participants with ASD in the fMRI response within foveal hMT+ (Fig. 2d–f) but not in the foveal region of EVC near the occipital pole (at the confluence of V1, V2, and V3; Fig. 2g–i) that provides input to area MT. At first, this finding may appear at odds with the notion of narrower top–down modulation in ASD, since top–down effects such as spatial attention are known to modulate responses in V1[48–53]. However, the magnitude of these modulatory effects varies greatly across different regions of visual cortex. In general, larger effects of top–down modulation have been observed in higher visual areas (like hMT+), vs. smaller effects at earlier stages[54]. Thus, in the current study, top–down modulation might be reflected to a greater degree in the fMRI responses within hMT+, as compared to those in EVC. Future studies that investigate other specialized, later stages of processing (e.g., responses to face stimuli in fusiform cortex) will be better positioned to address the specificity of our findings and determine (for example) whether weaker suppression in ASD is restricted to motion stimuli or is a general feature of higher-level visual processing.

Weaker neural suppression in ASD could be expected to have important consequences for sensory processing in one's daily life. A straightforward prediction is that reduced neural suppression would manifest in terms of increased sensory sensitivity. Our results bore this prediction out in a limited way; following correction for multiple comparisons, we found a non-significant trend toward a correlation between higher sensory sensitivity + avoiding scores and weaker fMRI suppression within foveal hMT+ across both participant groups (Fig. 3). Although this

observation was limited to the visual system, it is known that abnormal sensory phenomena can occur across modalities in ASD[1]. In addition to confirming or refuting the suggested relationship between visual suppression and sensory sensitivity, future research that examines neural suppression in different modalities (e.g., vision, somatosensation, and audition) may provide greater clarity regarding the link between abnormal neural suppression and sensory symptoms.

Our MRS measurements did not indicate a clear difference in signals related to inhibition or excitation between participants with ASD and NTs; MRS measurements of GABA+ and Glx in visual cortex did not differ between groups (Supplementary Figs. 2c and 3c) and were not correlated with fMRI or perceptual measures of suppression (Supplementary Fig. 2e, f). Our MRS results are in line with previous observations of normal GABA+ levels within visual cortex in people with autism[55–57] (but see ref. [8] for a more nuanced report). However, we cannot rule out the possibility that the mixed nature of the MRS signals (e.g., GABA plus co-edited macromolecules) could have masked subtle underlying group differences in the neurotransmitter signals of interest. Thus we cannot reach any strong conclusions regarding the role of GABA+ or Glx in visual cortex during spatial suppression among people with ASD.

We note that the individuals with ASD who participated in our study were generally high functioning, as reflected in their relatively high non-verbal IQ scores (Table 1). This may be a consequence of recruiting participants who were willing and able to take part in a demanding set of behavioral and neuroimaging experiments over the course of multiple days. We took care to ensure that the ASD and NT groups were well matched in terms of IQ, as previous studies have shown that individuals with higher IQ may show greater spatial suppression during motion discrimination[41–43]. Future studies may help clarify the extent to which weaker spatial suppression in ASD also applies to lower-functioning individuals on the spectrum.

## Methods

**Participants.** Our study included 28 young adult participants on the autism spectrum (18 males, 10 females), and 35 NT comparison participants (21 males, 14 females). Data from these participants with ASD[58–60] and NTs[12,58–61] were included in our recently published work. All participants were assessed by clinicians with extensive experience with ASD, under the supervision of a doctoral-level clinical psychologist who had achieved research reliability in the gold standard tools used to diagnose ASD. Diagnoses were confirmed through the ADOS-2[32], Autism Diagnostic Interview-Revised (ADI-R)[62], and clinical judgment using Diagnostic and Statistical Manual of Mental Disorders, 5th Edition criteria[63]. The following demographic factors did not differ significantly between the two participant groups: age, biological sex, non-verbal IQ (from the Wechsler Abbreviated Scale of Intelligence (WASI)[64], and handedness. Total scores on the Social Responsiveness Scale, 2nd Edition[65] were significantly higher among ASD participants. See Table 1 for demographic information and comparisons between groups. Individuals provided written informed consent prior to participation and were compensated $20 per hour. All procedures were approved by the Institutional Review Board at the University of Washington and conformed to the guidelines for research on human subjects from the Declaration of Helsinki.

Our inclusion criteria were as follows: age 18–30 years, non-verbal IQ >70, normal or corrected to normal visual acuity and no visual impairments, no impairment to sensory or motor functioning, no history of seizures or diagnosis of epilepsy, no neurological disease or history of serious head injury, no nicotine consumption in excess of 1 cigarette per day within the past 3 months, no use of illicit drugs within the past month, no consumption of alcohol within 3 days prior to MR scanning, and no conditions that would prevent safe and comfortable MR scanning (e.g., implanted medical devices, claustrophobia). In addition, individuals with ASD were not included if they had a change in their psychotropic medication within the past 6 months. NT individuals with a personal or family history of autism were not included. Two NT participants were taking prescribed antidepressants; excluding these two individuals did not qualitatively affect our results.

Participants who did not achieve criterion performance on catch trials in our behavioral task (one with ASD, two NTs; see Data analysis and statistics section below) were excluded from all analyses (behavioral data, functional MRI, and MRS). After exclusion, our final study sample consisted of 28 participants with

ASD and 35 NT participants. Demographic information from excluded participants are not included in Table 1. One participant with ASD was excluded from fMRI and MRS analyses (but not behavioral data) due to excessive head motion in the scanner. A summary of missing and excluded data is provided in Supplementary Table 1.

**Visual display and stimuli.** Our experimental apparatuses and stimuli have been described in our recent publications[12,61]. Visual experiments were conducted using three different display devices: (1) a ViewSonic PF790 CRT monitor (120 Hz) and Bits# stimulus processor (Cambridge Research Systems, Kent, UK) were used for all psychophysical experiments outside of the scanner. These stimuli were created and displayed in MATLAB (MathWorks, Natick, MA) and PsychToolbox 3[66–68]. Visual stimuli during our fMRI experiments were presented using either (2) an Epson Powerlite 7250 or (3) an Eiki LCXL100A projector (following an equipment failure; both at 60 Hz). These stimuli were created in MATLAB and presented using the Presentation software (Neurobehavioral Systems, Berkeley, CA). Viewing distance for all experiments was 66 cm, and luminance was linearized using a PR650 spectrophotometer (Photo Research, Chatsworth, CA).

Stimuli were sinusoidally modulated luminance gratings presented on a mean luminance background (Fig. 1a, b). In our psychophysical paradigm outside of the scanner, vertically oriented gratings drifted either left or right (drift rate = 4 cycles/s) within a circular aperture, which was blurred with a Gaussian envelope (SD = 0.21°). We used three different stimulus sizes: 0.84°, 1.7°, and 10° in diameter. The Michelson contrast of the gratings was either 3% (low) or 98% (high), and the spatial frequency was 1.2 cycles/°. Stimuli in the fMRI experiment differed from those in psychophysics as follows: diameter = 2° or 12°, spatial frequency = 1 cycle/°, Gaussian envelope SD = 0.25°.

**Psychophysics.** Our psychophysical paradigm, designed to measure spatial suppression, follows the methods of Foss-Feig and colleagues[4] and has recently been described[12,61]. In this task, participants were asked to discriminate the direction of motion (left or right) of a briefly presented drifting grating (Fig. 1c). Trials began with a shrinking circle fixation mark (850 ms) at the center of the screen, followed by a vertical grating, and then a response period (no time limit). Grating duration was adjusted across trials (range 6.7–333 ms) according to an adaptive (Psi) staircase procedure implemented within the Palamedes toolbox[69]. Correct responses tended to yield shorter durations on subsequent trials. In this way, grating duration was adjusted in order to find the briefest presentation for which the participant would perform with 80% accuracy. Each staircase was composed of 30 trials. Six independent staircases (3 sizes × 2 contrasts) were included in each run and were randomly interleaved across trials. Each run also included 10 catch trials (large, high-contrast gratings, 333 ms duration). These low-difficulty trials were intended to measure off-task performance. ASD participants showed slightly higher catch trial accuracy compared to NTs (ASD mean = 98.3%, SD = 2.8%, NT mean = 96.6%, SD = 4.7%; ANOVA, main effect of group, $F_{1,61} = 4.01$, $p = 0.0496$), consistent with the idea of narrower top–down modulation (e.g., spatial attention) in ASD. A total of four runs were included in each experimental session, which began with a set of examples and practice trials. Total task duration was approximately 30 min. Data were not obtained in the smallest stimulus size conditions for five participants in the ASD group and five NTs (for a summary of missing and excluded data, see Supplementary Table 1).

**Functional MRI.** Our fMRI paradigm was designed to measure spatial suppression and has also been described in a recent study from our group[12]. In this task, smaller (2° diameter) and larger (12°) drifting gratings were presented at the center of the screen in alternating 10 s blocks. Grating duration was 400 ms; inter-stimulus interval (ISI) was 225 ms. There were 16 gratings in each block, which drifted in 1 of the 8 possible directions (order randomized and counterbalanced). A single fMRI scanning run (4.2 min long) included a total of 25 blocks (13 smaller, 12 larger). Stimulus contrast was either 3% or 98% in separate runs. No baseline or rest blocks were included; stimuli appeared within the central 2° in all blocks. This paradigm not only allowed us to directly compare the change in the fMRI signal for larger vs. smaller stimuli but also prevented us from quantifying the responses to these two stimulus sizes independently from one another. Previous studies[70,71] have used this type of alternating block design to measure surround suppression in EVC using fMRI. We chose this paradigm to measure fMRI suppression in ASD for its simplicity and because it allowed us to easily exclude particular blocks from analysis (see below). Each participant completed 2–4 runs at each contrast level across 1 or 2 scanning sessions (some participants chose to end the experiment early, e.g., due to fatigue).

During fMRI, participants performed a color–shape conjunction task at fixation, responding to a green circle in a series of briefly presented colored shapes (e.g., blue square, green square, purple circle). Shapes (0.5° diameter) were presented at the center of the screen (i.e., on top of the gratings) for 66 ms every 1333 ms. Participants responded to the presentation of a green circle by pressing a button on an MR-compatible response pad (Current Designs, Philadelphia, PA). This task encouraged participants to keep their eyes and spatial attention fixed at the center of the screen. We also sought to emphasize bottom–up stimulus

processing of the grating stimuli by diverting attention toward the colored shapes in the fixation task.

Our fMRI experiment also included two functional localizer scans, which were used to identify regions of interest (ROIs). The first localizer was designed to identify the motion-selective brain area known as human MT complex (hMT+; Supplementary Fig. 1a). We refer to this area as hMT+ to indicate that we did not attempt to differentiate area MT and the medial superior temporal area[72]. This localizer consisted of alternating 10 s blocks of drifting and static gratings (2° diameter, 15% contrast; Supplementary Fig. 1b). There were 25 blocks in total (13 static, 12 drifting). Grating duration was 400 ms with a 225 ms ISI. The second localizer scan was used to identify voxels with retinotopic selectivity for the central 2°. Using a differential localizer approach[53,73] allowed us to identify voxels that responded more strongly to stimuli in the center vs. surrounding portion of the screen. This scan consisted of alternating 10 s blocks of phase-reversing checkerboards (8 Hz; 100% contrast) within the central 2° or within an annular region from 2° to 12° eccentricity (Supplementary Fig. 1c, d). There were 16 blocks in the second localizer scan (8 center, 8 annulus). Rest blocks were not included during either localizer. Participants performed the same fixation task as in the main fMRI experiment during both localizers. One run of each localizer type was included in each scanning session.

Prior to MR scanning, participants completed a 30-min mock scanning session in which they were introduced to the scanner sounds and practiced lying still in a simulated scanner environment. Participants wore a 3D Guidance trakSTAR motion sensor (Ascension Technology Corp., Shelburne, VT) mounted on a headband. Visual feedback on head motion was given using MoTrak 1.0.3.4 software (Psychology Software Tools, Inc., Sharpsburg, PA). Participants were instructed to keep a small dot representing their head position within a bullseye target region. The mock scanning session also included a practice session for the fMRI fixation task, with examples of the colored shape stimuli, as well as the gratings and checkerboards from each of the fMRI scans.

During the fMRI experiment, we examined fMRI data for head motion as soon as they were acquired. Immediately after each run, data were transferred off the scanner and motion correction was performed using BrainVoyager (see below). Runs in which substantial and sudden head movements (e.g., >2 mm across 2–4 s) were detected were excluded and repeated within the same scanning session. In such cases, participants were given feedback and coached to remain as still as possible during the subsequent run.

MR data were acquired on a Philips 3 tesla scanner. Each scanning session began with a $T_1$-weighted anatomical scan (1 mm isotropic resolution), followed by whole-brain gradient echo fMRI (3 mm isotropic resolution, 30 oblique-axial slices with a 0.5 mm gap, 2 s TR, 25 ms echo time [TE], 79° flip angle, anterior–posterior phase encoding direction). A single run with the opposite phase encoding direction (posterior–anterior; 3 TRs) was also acquired during each scanning session to facilitate geometric distortion compensation.

**MR spectroscopy.** We conducted a $^1$H MRS experiment designed to measure GABA+ within particular brain regions. We refer to this metric as GABA+ to indicate that it reflects GABA plus co-edited macromolecules, which are not differentiated by this approach[74]. Our methods were described in our recent publications[12,61]. Briefly, we used a MEGA-PRESS sequence[75] to obtain edited MRS data within a 3 cm isotropic voxel (320 averages, 2 s TR, 68 ms TE, 2048 spectral data points, 2 kHz spectral width, 1.4 kHz refocusing pulse, VAPOR water suppression). Fourteen-ms editing pulses were applied at 1.9 ppm (on) or 7.5 ppm (off) during alternating acquisitions within a 16-step phase cycle. The duration of a single MRS run was approximately 11 min. This MRS method provides a static measure of GABA levels, which may better reflect inhibitory tone rather than dynamic inhibition per se. An in-session anatomical scan (as above) was acquired prior to the MRS runs. In order to achieve adequate signal to noise, we measured GABA+ within a relatively large (27 mm³) brain region, which necessarily limits the spatial precision of our data.

We acquired MRS data in two regions of visual cortex: (1) the region of the lateral occipital lobe surrounding area hMT+ (Supplementary Fig. 2a, b; acquired bilaterally in separate runs) and (2) a region of EVC (Supplementary Fig. 3a, b) in the medial occipital lobe aligned parallel and positioned dorsal to the cerebellar tentorium. The hMT+ MRS voxel was placed based on an in-session functional localizer fMRI scan designed to identify area hMT+ (as above, but with duration = 195 s, TR = 3 s, resolution = 3 × 3 × 5 mm, 14 slices with 0.5 mm gap). The hMT+ region was identified online at the scanner using Philips iViewBOLD to identify voxels in the lateral occipital lobe that responded significantly more strongly to moving vs. static gratings ($t \geq 3.0$). Functional localizer data during the MRS experiment were acquired prior to the anatomical scan and subsequent MRS runs, in order to minimize the effect of frequency drift during MRS caused by gradient heating during fMRI. EVC MRS voxels were positioned according to anatomical landmarks. We acquired MRS scans in a fixed order (left hMT+ first, EVC in the middle, right hMT+ last), to ensure that any effects of gradient heating would be equivalent within a given voxel across all participants and groups. We did not observe any difference in water frequency drift (SD in Hz across the scan) when comparing the first (left hMT+; mean = 1.09 Hz) and last (right hMT+; mean = 0.99 Hz) MRS scans ($t_{61} = 1.17$, $p = 0.25$), thus we do not believe that there were large systematic differences in data quality between these two runs due

to gradient heating. In order to maximize compliance during MRS, participants watched a theatrical film of their choice to reduce boredom and fatigue, as we have found this can help participants better tolerate long (>1 h) scanning sessions. Although we assume that GABA+ values measured with MEGA-PRESS at 3 T are relatively stable within individuals[76–78], differences in metabolite levels between participants due to varying visual stimulation may be a source of unaccounted variance in our MRS data.

**Computational modeling.** We applied the normalization model developed by Reynolds and Heeger[35] to describe motion duration threshold data, as in our previous work[12]. A model diagram is provided in Fig. 4. The model can be summarized by the equation:

$$R = \frac{E \times M}{S \times S_g + \sigma} \qquad (1)$$

where $R$ is the predicted model response (in arbitrary units), $E$ is the feed-forward excitatory drive (a function of the visual stimulus strength), $M$ is the top–down gain modulation field (a parameter that scales $E$ and reflects how stimulus processing is modulated by top–down factors). $S$ is the suppressive drive (which depends on the excitatory drive [$E$], but is spatially broader, representing the contribution of a broad normalization pool), $S_g$ is the suppressive gain (a scaling factor for the suppressive drive, representing the strength of normalization), and $\sigma$ is the semi-saturation constant (a small number that prevents the function from being undefined when the value of $S$ is zero). We note that the spatial selectivity of $E$ is determined by a Gaussian function with a width parameter $x_{w\_e}$ (Supplementary Eq. 2); we refer to this parameter as the width of the excitatory SF (akin to a neural receptive field). This model differs from that in our previous work[12] in two important ways: (1) the inclusion of the suppressive gain parameter $S_g$ and (2) the addition of the top–down modulation parameter $M$. Please see Supplementary Methods for full model details and Supplementary Table 2 for all parameter values.

To predict duration thresholds from model responses, we assumed an inverse relationship between response magnitude and duration thresholds, such that:

$$T = \frac{C}{R_{peak}} \qquad (2)$$

where $T$ is the predicted model threshold (in arbitrary units), $C$ is the criterion response value needed to reach a perceptual judgment (i.e., leftward vs. rightward motion direction)[79], and $R_{peak}$ is the peak region of the predicted model response from Eq. (1). This inverse relationship between threshold and response is consistent with previous models of motion duration thresholds[3,38,80] and with electrophysiological data from nonhuman primates recorded in area MT during a comparable motion discrimination task[14].

We compared three different model variants (Fig. 5; Supplementary Fig. 5) to determine which might best match our observation of weaker spatial suppression during motion discrimination in people with ASD vs. NTs (Fig. 2a–c). First, we considered the effect of weaker normalization (i.e., a 25% reduction in the suppressive gain term $S_g$, from 1 to 0.75) on duration thresholds predicted by the model (see Fig. 4, orange box; Fig. 5a–c). Using the normalization model, Rosenberg and colleagues[2] proposed that such a reduction in suppressive gain might account for lower motion duration thresholds in ASD, as reported by Foss-Feig and colleagues[4]. We note that reducing the suppressive gain factor necessarily shrinks the effective size of the suppressive drive in both the spatial and orientation dimensions, as scaling down a two-dimensional Gaussian brings all values closer to zero multiplicatively. Next, we examined the effect of larger excitatory SFs in the model (25% increase in $x_{w\_e}$, the width of the Gaussian function that determines the spatial selectivity of $E$; Fig. 4, magenta box; Fig. 5d–f). Schauder and colleagues[3] found that larger SFs were able to explain their observation of higher duration thresholds in people with ASD vs. NTs. Finally, we examined whether changing the width of the top–down modulation parameter $M$ might affect spatial suppression as predicted by the normalization model (Fig. 4, cyan box; Fig. 5g–i). In particular, we used a narrower width for the Gaussian parameter that determines the spatial selectivity of $M$ in our model (6 vs. 14 arbitrary units). This is consistent with the idea of narrower top–down processing (e.g., spatial attention) during visual perception in ASD, as suggested by previous experimental findings[5]. These three model variants were compared for a qualitative match to the pattern of motion duration thresholds we observed in people with ASD vs. NTs (Fig. 2a–c). Although other model variants have been successfully used to describe motion duration threshold data (e.g., divisive models with different contrast sensitivity for excitation and suppression[3,38]), normalization models such as the one applied here have also proven effective[2,12]. We chose to use a normalization model here in order to directly test hypotheses advanced in earlier theoretical work (e.g., the idea that ASD is associated with weaker normalization[2]) against our current experimental data. Additional modeling details are provided in Supplementary Methods (Supplementary Table 2), including an explanation of how our model implementations compare to previous work.

**Clinical measures.** Clinical and cognitive assessments were conducted by clinicians with expertise in the evaluation of individuals with neurodevelopmental disorders and who achieved research reliability on the ADOS-2 and ADI-R; autism

diagnoses were confirmed by a trained doctorate-level clinical psychologist using all available information. Overall autism symptom severity was estimated using the ADOS-2 total comparison score[32]. To examine sensory sensitivity and sensory avoidance, we used the corresponding domains from the Sensory Profile[33,34]. Because these two subscales were highly correlated in our sample ($r_{60} = 0.82$, $p = 3 \times 10^{-16}$), we summed them in order to treat them as a single, combined measure of sensory dysfunction.

**Data analysis and statistics**. Psychophysical data were analyzed in MATLAB using the Palamedes toolbox[81]. Duration thresholds for motion direction discrimination were calculated by fitting a Weibull function to the data from each individual staircase. Guess rate and lapse rate were fixed at 50% and 4%, respectively. Thresholds were calculated from the fit psychometric function as the duration value where the participant performed with 80% accuracy. Threshold values <0 or >500 ms were excluded (pre-defined criteria); a total of 3 thresholds were excluded in this way across all data sets. Catch trial accuracy was assessed separately from the staircase data to examine off-task performance. Participants with <80% accuracy across all 40 catch trials were excluded (pre-defined criterion) from all further analyses, including fMRI and MRS. One participant with ASD and two NTs were excluded in this manner (for a summary of missing and excluded data, see Supplementary Table 1).

To quantify the effect of increasing stimulus size on motion discrimination performance (i.e., suppression), size indices were calculated using an established method[4,12], according to Eq. (3):

$$\text{Size index} = \log_{10}(\text{medium}) - \log_{10}(\text{big}) \qquad (3)$$

where medium indicates the threshold in the medium size condition (1.7° diameter) for a particular contrast level (either 98% or 3%) and big indicates the threshold for the big size condition (10° diameter) for the same contrast. More negative size indices indicate stronger suppression (i.e., a bigger increase in motion duration thresholds with increasing stimulus size).

Functional MRI data were processed in BrainVoyager (Brain Innovation, Maastricht, Netherlands). This included motion correction, distortion compensation, high-pass filtering (>2 cycles/run), and alignment to the in-session anatomy. Spatial smoothing and normalization to a canonical template were not performed; all analyses were based on within-subject ROIs. ROIs were defined in each hemisphere in the space of the functional data using a standard correlational analysis[12,53], taking the top 20 most significant voxels with an initial threshold of $p < 0.05$ (Bonferroni corrected for multiple comparisons). In a few cases, there were not 20 voxels that satisfied this threshold, thus the threshold was relaxed to include 20 contiguous voxels. The same number of voxels (40 across both hemispheres) are included in each ROI for each participant. All ROIs satisfied a minimum threshold of $p < 0.006$ (one-tailed, uncorrected). EVC ROIs in four participants with ASD and one NT did not satisfy this post hoc criterion; these participants were excluded from EVC fMRI analyses (for a summary of missing and excluded data, see Supplementary Table 1).

ROI location was verified through visualization on an inflated cortical white matter surface model. Bilateral ROIs were defined for area hMT+ in the lateral occipital lobe (Supplementary Figs. 1 and 2) from the motion vs. static functional localizer data and for EVC near the occipital pole from the center vs. surround functional localizer data. ROIs in hMT+ were further refined by finding voxels within hMT+ that additionally showed significant retinotopic selectivity for the central 2° in the center vs. surround functional localizer (one-tailed $p < 0.05$; Supplementary Fig. 1c, d). Thus the hMT+ ROIs were identified from the intersection of voxels in the lateral occipital lobe showing selectivity for motion > static and center > surround. Center-selective regions within hMT+ could not be identified in three participants with ASD and ten NTs, who were thus excluded from hMT+ fMRI analyses (pre-defined criterion; for a summary of missing and excluded data, see Supplementary Table 1). Identifying center-selective regions within hMT+ was critical for observing the spatial suppression effect of interest; when stimulus size increases, hMT+ voxels with a peripheral retinotopic bias will respond more strongly. However, the center vs. surround functional localizer stimuli (flickering checkerboards) were designed with EVC in mind and may not have been optimal for identifying foveal regions of hMT+. Future studies may benefit from using localizer stimuli better suited for eliciting strong responses in hMT+ (e.g., center and surround defined by drifting gratings or dots).

We extracted fMRI data from each ROI for further analyses in MATLAB using BVQXTools. We examined how fMRI signals within each ROI changed when the stimulus size increased. Within each experimental condition (i.e., high and low stimulus contrast), ROI time series data were divided into epochs spanning 4 s before to 12 s after the stimuli changed from smaller to larger (event-related time = 0 s). Response baseline was calculated by averaging the signal from 0 to 4 s before the size change across all epochs. We converted the data to percentage of signal change by subtracting and then dividing by the baseline value and then multiplying by 100. To compute an average response time course in each condition for each participant, we took the mean signal for each time point across all epochs, hemispheres, and fMRI runs. The magnitude of the fMRI response to the increase in stimulus size was calculated as the average signal from 8 to 12 s after the size increase (the time period when suppression was maximal).

MRS data were analyzed in the Gannet 2.0 Toolbox[82] within MATLAB. Data were processed using the toolbox-standard approach, including automated frequency and phase correction, artifact rejection (frequency correction >3 SD above the mean), and 3 Hz exponential line broadening. To calculate the concentration of GABA+, we fit a Gaussian to the peak in the MEGA-PRESS spectrum at 3 ppm (Supplementary Fig. 7). We refer to this value as GABA+ to indicate that it reflects GABA plus co-edited macromolecules, which are not differentiated by this method. Likewise, the Glx peak at 3.75 ppm was fit with a double Gaussian. The area under the fit curve served as a measure of the metabolite level. GABA+ and Glx were each scaled relative to water; the unsuppressed water peak was fit with a mixed Gaussian–Lorentzian. Tissue correction was performed for GABA+ based on the proportion of gray matter, white matter, and cerebrospinal fluid within each MRS voxel based on the relaxation properties of different tissue types[83,84], and assuming twice the concentration of GABA+ in gray vs. white matter, using an established method[85]. Tissue fraction values within each MRS voxel were obtained by segmenting the $T_1$ anatomical scan using SPM8[86]. There is currently no standard method for tissue correction for Glx. Instead, we performed a series of control analyses to explore the contributions of different tissue types and the measured water reference signal to our Glx measurements (see Supplementary Notes). Concentrations for GABA+ and Glx are reported in institutional units (i.u.). We collected all psychophysical, fMRI, and MRS data within a 2-week time period for each participant; previous studies suggest that GABA+ values are fairly stable over this time period[76–78]. One participant with ASD was excluded from MRS analyses due to excessive head motion, as evidenced by large water frequency shifts across time (post hoc assessment; for a summary of missing and excluded data, see Supplementary Table 1).

Statistical analyses were performed in MATLAB. Normality and homogeneity of variance were assessed by visual inspection of the data. Group differences were assessed using mixed repeated-measures ANOVAs. Participants were modeled as a random effect and nested within groups. Stimulus size was modeled as a continuous variable. Stimulus size and contrast were treated as within-subjects factors. Reported $r$ values are Pearson's correlation coefficients; associated $p$ values were determined using non-parametric two-tailed permutation tests, unless otherwise noted. Here we randomly shuffled the data being correlated across participants (without replacement) in each of the 10,000 iterations; $p$ values were calculated as the proportion of shuffled samples with $r$ values more extreme than the observed $r$ value. Bonferroni correction was used to adjust $p$ values for multiple comparisons. When comparing the proportion of left- and right-handed participants between groups (Table 1), Yates's correction was used to adjust the $X^2$ value for low expected counts.

**Reporting summary**. Further information on research design is available in the Nature Research Reporting Summary linked to this article.

## Data availability
Data from this study are available at https://nda.nih.gov/edit_collection.html?id=2266. The following figures have associated raw data: Figs. 2 and 3, Supplementary Figs. 2, 3, 6, and 7.

## Code availability
MATLAB code for our implementation of the normalization model is available at https://github.com/mpschallmo/WeakerNeuralSuppressionAutism.

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

## Acknowledgements
We thank Brenna Boyd, Judy Han, Ly Nguyen, Heena Panjwani, Micah Pepper, Meaghan Thompson, Anne Wolken, and the UW Diagnostic Imaging Center for help with recruitment and/or data collection. We thank Geoffrey M. Boynton for providing the original MATLAB functions for the computational model and Mark Mikkelsen for help with MRS quantification. This work was supported by funding from the National Institutes of Health (F32 EY025121 to M.-P.S., R01 MH106520 to S.O.M., T32 EY00703). This work applies tools developed under NIH grants R01 MH098228, R01 EB016089, and P41 EB015909; R.A.E.E. also receives support from these grants.

## Author contributions
M.-P.S., A.V.F., R.A.B., and S.O.M. designed the study. M.-P.S., A.V.F., R.A.E.E., R.A.B., and S.O.M. developed the methods. M.-P.S., T.K., A.M.K., R.M., and J.G. collected the data. M.-P.S., T.K., A.M.K., R.M., A.V.F., J.G., and S.O.M. analyzed the data. M.-P.S. and S.O.M. wrote the paper. All authors edited the paper.

## Competing interests
The authors declare no competing interests.
