## [Peer Review File · Nature Communications]

Reviewers' Comments:

Reviewer #1:

Remarks to the Author:

Schallmo and colleagues.

This is a very interesting set of experiments and an excellent write-up from Schallmo and colleagues. It addresses an interesting and unresolving question of longstanding interest in the literature that hypothesises an excitation/inhibition imbalance as a fundamental underlying motif in autism. This has been difficult to test because of the lack of suitable animal models for the disorder, and the limitations of non-invasive testing in humans in addressing questions about neurotransmitters. Schallmo and colleagues go some way to circumventing these issues by cleverly leveraging a visual neuroscience paradigm whose effects at a circuit level are well known through intensive psychophysical and physiological study, combining this with non-invasive MR spectroscopy to measure signals associated with excitatory and inhibitory neurotransmission. Their findings are consistent with reduced inhibition in some (but not all) areas of visual cortex, that is not attributable to altered GABA levels but instead is consistent with a modification of top-down gain control predicted by an influential theory of visual circuits.

I have some minor suggestions for improving the manuscript as follows:

1. Autism/ASD is a highly heterogeneous disorder so some comment on the representativeness of the sample and generalisation (or not) to an ASD population would be worthwhile in the discussion. Table 1 suggests that the sample were in general functioning at a relatively high level.
2. There's a minor inconsistency between the methods which say the original ADOS was used and Table 1 which suggests the revised ADOS-2 was used.
3. Figure 1 legend needs to detail the statistical threshold used for the '*' in panels C/F/I
4. The difference between fMRI suppression in EVC versus MT+ is striking and interesting, as it potentially rules out a generalised mechanism (albeit there could be a stimulus-specific mechanism). I don't think the authors carried out a region x suppressive effect interaction analysis which would be helpful
5. The correlation between fMRI suppression and sensory sensitivity is interesting, but it was non-significant (figure 4). I suggest the mention of this correlation in the Discussion (at the very end) is therefore accompanied by mention of this fact, while leaving the possibility open of further work to confirm or refute this possible association.
6. I couldn't see any mention of actual performance on the fMRI fixation task/catch trials beyond the exclusion threshold of 80% correct performance mentioned in the Data Analysis. It would be helpful to include a description of whether there were any differences in performance on the catch trials between the group – especially as an attentional hypothesis is considered for the differences in suppression observed between groups.

Reviewer #2:

Remarks to the Author:

Schallmo and colleagues compare spatial suppression in individuals with and without autism using an

impressive combination of psychophysics, modeling, and neuroimaging. They present behavioral evidence for weaker spatial suppression in autism, particularly at high stimulus contrasts. This behavioral difference is reflected in weaker suppression of BOLD signals in hMT+ by large as compared with medium stimuli, suggesting a neural correlate to their behavioral effect. They implement a divisive normalization model to describe these results, which suggests that a reduction in the spatial extent of top-down gain best recapitulates differences in autism. They also measure neurotransmitter levels in hMT+ using MRS, but find that levels of GABA and Glx do not track with the task in either group and do not differ between groups.

This paper is a tour de force, taking a comprehensive approach to characterizing an impressively large set of autism data from multiple angles. They present a large, well-characterized sample of participants, who are matched on psychometric measures (age, gender, and IQ). Further, the question they tackle – whether a visual paradigm like spatial suppression can provide insight into the computational principles of autism neurobiology – is important and timely.

However, the conclusions of the paper, as they stand, are not well supported by the results. Further, behavioral and computational modeling studies of the same paradigm tested in this paper have already been published (Foss-Fieg 2013, Schauder et al, 2017, Rosenberg et al., 2015) and reach different conclusions than are presented here. It is important that the authors address these previous findings upfront, and explain why their conclusions differ and their results are justified. That said, this paper does a far more comprehensive job of exploring spatial suppression in autism than these previous papers by including a large sample size and a neuroimaging component – and I believe that this contribution will justify its ultimate publication. Specific comments follow below.

Major comments:

1. I am perplexed by the framing of this paper in the context of testing the hypothesis that “ASD may result from a pervasive reduction of neural inhibition”. As the authors note, recent models suggest that spatial suppression effects with larger stimuli are best explained by a withdrawal of excitatory input compared with increased lateral inhibition (Sato et al. 2016, Liu et al., 2018). Perhaps consistent with these models, the authors find that GABA levels in hMT+ do not predict the strength of spatial suppression in control individuals, also suggesting that this paradigm is not a good proxy for inhibition in the brain. If the paradigm used in this paper is not a probe of inhibition in the brain – why is this paper framed as a test of inhibition in autism?
2. The computational modeling results, on the other hand, take an unbiased approach to describing the robust behavioral and fMRI evidence presented for weaker spatial suppression in autism. These results follow on from a recent application of the divisive normalization model to other data presenting weaker spatial suppression results in autism (Rosenberg et al., 2015). But Rosenberg et al. reach a different conclusion than the authors present here. The differences between these two papers are not well laid out. The authors should present this previous work upfront, describe the parameters of the divisive normalization model used in Rosenberg et al., 2015 vs in the present work, and justify why the current conclusions presented here differ from those found in Rosenberg et al., 2015.
3. For example, Rosenberg et al., 2015 found that a 75% reduction of the suppressive field gain term (relative to controls) describes weaker spatial suppression in autism at high contrasts, but not changes to excitatory gain. Here, the authors chose to test a 25% reduction of the suppressive field gain term relative to controls and reach a different conclusion. Why?
4. It is difficult to understand the potential impact of the many degrees of freedom in this model on the presented results, especially when authors only present results after tweaking specific parameters to a specific level. To more fully understand the specificity of the presented results to specific levels and parameters, please present a full simulation landscape plotting the size of the suppression index

(big – med) as a function of +/- alterations in suppressive gain vs. excitatory spatial filters and suppressive gain vs. top-down gain field size.

5. Given that neither GABA nor Glx levels predict task performance in either group (controls or autists), it is difficult for the authors to present their MRS data as strong evidence as to the neural basis of their behavioral effects on the tested task. There are many reasons for a lack of correlation between MRS measurements and task performance – ranging from the true dependence of this task on inhibition (as raised above) to the sensitivity of MRS measurements to detect such a dependence in the context of this task and scan protocol. Given that no relationship was observed in this study between MRS measurements and task performance in controls, this aspect of the paper is inconclusive.

6. The authors have strong, compelling behavioral effects suggesting that spatial suppression is altered in autism at high contrasts. However, this paradigm has previously been explored in autism in previous studies with mixed results (Schauder et al., 2017, Foss-Fieg et al., 2013, etc.). How did these previous results motivate the current work, and how should we interpret the conclusions of these papers? These issues should be discussed in the introduction of the paper, as the previous literature on this paradigm in autism is important context for the reader in interpreting the current results.

7. The strong behavioral evidence for weaker spatial suppression in autism is reflected in reduced spatial suppression in the BOLD Signal in hMT+. This suggests a brain-behavior link. Does the degree of spatial suppression in behavior predict that in fMRI?

8. A large amount of data that was lost in defining foveal hMT+ based on the intersection of a motion localizer and a center-surround checkerboard (10 control participants). Is this because the phase-reversing checkerboards do not optimally drive hMT+? Regardless, given that this definition was apparently not optimal for defining foveal hMT+, what do the results look like when constraining the analysis to hMT as defined by a traditional motion localizer?

9. From a brief study of the references cited in the first paragraph of the introduction, I am under the impression that the authors are dramatically oversimplifying the “excitation/inhibition” literature in autism. The results cited in mice and humans often pertain to specific regions of the brain – this is not necessarily evidence for “widespread reduction in inhibition”. Is the theory that certain regions of the brain are affected by reduced neural inhibition in autism? Certain circuits? Is it always reduced inhibition? Recent reviews of the E/I literature in autism and psychiatric conditions highlight the complexities of this theory, which are not captured in the current introduction. Further, given that the paradigm tested here is not really a test of inhibition (see comments above), the current introduction does not seem to accurately introduce the hypotheses tested in this paper relating to a specific neural computation, divisive normalization, in specific region of the autistic brain, hMT+/V1.

Minor comments:

1. The ANOVA reported in the behavioral results section - are size and contrast both included as within-subject variables?

2. Model - Is the spatial extent of inhibition also impacted by altering the amplitude of the inhibitory field in the current model?

3. fMRI - How big are the bilateral hMT+ ROIs, and are the two groups matched in size (voxel #)?

4. fMRI - the suppression effect is measured by taking BOLD response in foveal hMT+ to large - the preceding small stimuli. Results show weaker suppression in autism given this baseline. How do baseline responses (to the preceding small stimuli) compare between the groups?

5. MRS/fMRI - No details are given regarding MRS data quality (field drift, spectral line width, SNR, etc.) or fMRI data quality (in-scanner motion), as compared across groups.

Reviewer #3:

Remarks to the Author:

Review of Nature Comms.19-17709 Weaker Neural Suppression in Autism

The authors present a compelling multimodal study of fMRI and MRS in a well-grounded and theoretically-principled experimental design. They then invoked a computation model to account for observed data.

Consistent with prior studies of cortical GABA estimation in visual cortex (e.g. Gaetz et al.), no differences from typical levels are observed in ASD in this study, despite weaker neural suppression (measured behaviorally and by fMRI). They introduce a computational model and propose that weaker neural suppression could be attributable to differences in top-down processing, which is certainly interesting. A more trivial explanation could, however, lie in either the imprecision of GABA estimation per se, or the lack of specificity to the nature of GABA being estimated (since voxels are rather large compared with cellular compartments), leaving most to interpret such MRS-based GABA estimates (at best) as indices of inhibitory tone. While that could be addressed through slightly more conservative discussion of the interpretation of the findings, perhaps a more compelling experiment would be to attempt analogs in other cortices (if conceivable) where similarly-obtained GABA estimates have consistently shown deficits in ASD (e.g. sensorimotor, Gaetz et al., Puts et al. or auditory (Rojas et al., Gaetz et al., Port et al.)).

Minor points –

it is unfortunate that the NT group did not also undergo ADOS observation – how sure are we that the controls exhibit no ASD-tendencies ?

The proportion of males and females is similar in the groups, which is commendable. Is there sufficient sample to estimate a sex effect in the neural suppression or GABA levels (or both) ?

NVIQ (although not different between groups) seem rather high to be broadly representative.

It seems the GABA single voxel was placed according to fMRI results – necessitating the EPI run to precede the MRS – did the authors encounter any (very typical) issues with eddy current heating (and subsequently cooling) leading to field drift across the (11-minute) MRS acquisition ? Were the sequence of voxel placements randomized across subjects ? They argue that the fMRI preceded the anatomic scan too (was that sufficient to prevent field drift ? this is easily ascertained since 3 MRS acquisitions were performed – was the center frequency constant at the beginning and end of each acquisition) ?

Did GABA levels drift during the course of the fMRI stimulation paradigm ? While I am sure GABA and fMRI were not interleaved acquisitions (although this might be an interesting direction), was GABA estimation performed both before and after the visual paradigm (at least for the a priori selected EVC voxel placement) ?

Similarly it seems strange to allow the participants to watch different movies during GABA estimation in visual cortex – presumably GABA levels are approximately constant and insensitive to the nature of visual stimulation, but I would have thought no stimulation (or at least a constant level of luminance/contrast/dynamics would have been preferred).

What is the nature of the small peak at ~3.2ppm? It is excluded from analysis. Can the authors be

sure it is not attributable to GABA? The side-band of a GABA triplet might be expected to be about .2ppm apart I suppose. Or is it just an irrelevant metabolite that should indeed be discarded ? Does it contaminate the GABA estimates (i.e. should it be fitted out separately) ? On that point, it would probably be helpful to see the individual fits as a stacked plot. GANNET 2.0 fitting is extremely sensitive to any baseline fluctuation around 3-3.5ppm (since it fits a "Gaussian plus a line").

Reviewers' comments:

Reviewer #1 (Remarks to the Author):

Schallmo and colleagues.

This is a very interesting set of experiments and an excellent write-up from Schallmo and colleagues. It addresses an interesting and unresolving question of longstanding interest in the literature that hypothesises an excitation/inhibition imbalance as a fundamental underlying motif in autism. This has been difficult to test because of the lack of suitable animal models for the disorder, and the limitations of non-invasive testing in humans in addressing questions about neurotransmitters. Schallmo and colleagues go some way to circumventing these issues by cleverly leveraging a visual neuroscience paradigm whose effects at a circuit level are well known through intensive psychophysical and physiological study, combining this with non-invasive MR spectroscopy to measure signals associated with excitatory and inhibitory neurotransmission. Their findings are consistent with reduced inhibition in some (but not all) areas of visual cortex, that is not attributable to altered GABA levels but instead is consistent with a modification of top-down gain control predicted by an influential theory of visual circuits.

I have some minor suggestions for improving the manuscript as follows:

1. Autism/ASD is a highly heterogeneous disorder so some comment on the representativeness of the sample and generalisation (or not) to an ASD population would be worthwhile in the discussion. Table 1 suggests that the sample were in general functioning at a relatively high level.

The reviewer is right to point out that our sample of ASD participants was generally high functioning (i.e., normal non-verbal IQ scores). This is a somewhat necessary consequence of recruiting subjects who are able and willing to participate in a demanding set of psychophysical and MR experiments over the course of multiple days. We have added comments to this effect in the Discussion.

2. There's a minor inconsistency between the methods which say the original ADOS was used and Table 1 which suggests the revised ADOS-2 was used.

This has been corrected to reflect that ADOS-2 was used.

3. Figure 1 legend needs to detail the statistical threshold used for the '*' in panels C/F/I

*We have added a note to the legend of Figure 2 (the first results figure, as well as Supplemental Figure 5) to clarify that the * indicates a significant result at $p < 0.05$.*

4. The difference between fMRI suppression in EVC versus MT+ is striking and interesting, as it potentially rules out a generalised mechanism (albeit there could be a stimulus-specific mechanism). I don't think the authors carried out a region x suppressive effect interaction analysis which would be helpful

The reviewer is correct that we did not assess a group x brain area interaction in the fMRI results. We had treated these areas separately because they showed very different patterns of fMRI responses in our previous study of NTs (Schallmo, 2018), and we expected to see similarly divergent patterns here. Assessing this interaction now, to our surprise we find it is not statistically significant ($F_{1,46} = 0.13$, $p = 0.7$). We speculate that this may be the case because the ASD group shows numerically (but not statistically) weaker suppression for high (but not low) contrast stimuli in EVC, as compared to NTs. This trend seems to push the results of the omnibus ANOVA (for MT & EVC) toward an overall group difference (main effect of group; $F_{1,59} = 2.53$, $p = 0.117$). Alternatively, this may simply reflect the reduced statistical power of the 3-factor (group, stimulus contrast, brain area) ANOVA.

5. The correlation between fMRI suppression and sensory sensitivity is interesting, but it was non-significant (figure 4). I suggest the mention of this correlation in the Discussion (at the very end) is therefore accompanied by mention of this fact, while leaving the possibility open of further work to confirm or refute this possible association.

We now clarify at the end of the Discussion that this correlation showed a non-significant trend after correction for multiple comparisons, and note the need for confirmation in future studies.

6. I couldn't see any mention of actual performance on the fMRI fixation task/catch trials beyond the exclusion threshold of 80% correct performance mentioned in the Data Analysis. It would be helpful to include a description of whether there were any differences in performance on the catch trials between the group – especially as an attentional hypothesis is considered for the differences in suppression observed between groups.

Consistent with the reviewer's suggestion, we find that ASD participants showed slightly higher catch trial accuracy during psychophysics as compared to NTs (ASD mean = 98.3%, $SD = 2.8\%$, NT mean = 96.6%, $SD = 4.7\%$; ANOVA, main effect of group, $F_{1,61} = 4.01$, $p = 0.0496$). Hit rates in the fMRI fixation task were also higher among ASD participants, but this difference was not statistically significant (ASD mean = 95.0%, $SD = 7.0\%$, NT mean = 93.9%, $SD = 8.1\%$; ANOVA, main effect of group, $F_{1,61} = 0.31$, $p = 0.6$).

These findings are now reported alongside the description of these measures in the Methods, and we note that superior catch trial performance is consistent with the idea of narrower top-down modulation (e.g., attention) in ASD, as the reviewer suggests.

Reviewer #2 (Remarks to the Author):

Schallmo and colleagues compare spatial suppression in individuals with and without autism using an impressive combination of psychophysics, modeling, and neuroimaging. They present behavioral evidence for weaker spatial suppression in autism, particularly at high stimulus contrasts. This behavioral difference is reflected in weaker suppression of BOLD signals in hMT+ by large as compared with medium stimuli, suggesting a neural correlate to their behavioral effect. They implement a divisive normalization model to describe these results, which suggests that a reduction in the spatial extent of top-down gain best recapitulates differences in autism. They also measure neurotransmitter levels in

hMT+ using MRS, but find that levels of GABA and Glx do not track with the task in either group and do not differ between groups.

This paper is a tour de force, taking a comprehensive approach to characterizing an impressively large set of autism data from multiple angles. They present a large, well-characterized sample of participants, who are matched on psychometric measures (age, gender, and IQ). Further, the question they tackle – whether a visual paradigm like spatial suppression can provide insight into the computational principles of autism neurobiology – is important and timely.

However, the conclusions of the paper, as they stand, are not well supported by the results. Further, behavioral and computational modeling studies of the same paradigm tested in this paper have already been published (Foss-Fieg 2013, Schauder et al, 2017, Rosenberg et al., 2015) and reach different conclusions than are presented here. It is important that the authors address these previous findings upfront, and explain why their conclusions differ and their results are justified. That said, this paper does a far more comprehensive job of exploring spatial suppression in autism than these previous papers by including a large sample size and a neuroimaging component – and I believe that this contribution will justify its ultimate publication. Specific comments follow below.

Major comments:

1. I am perplexed by the framing of this paper in the context of testing the hypothesis that “ASD may result from a pervasive reduction of neural inhibition”. As the authors note, recent models suggest that spatial suppression effects with larger stimuli are best explained by a withdrawal of excitatory input compared with increased lateral inhibition (Sato et al. 2016, Liu et al., 2018). Perhaps consistent with these models, the authors find that GABA levels in hMT+ do not predict the strength of spatial suppression in control individuals, also suggesting that this paradigm is not a good proxy for inhibition in the brain. If the paradigm used in this paper is not a probe of inhibition in the brain – why is this paper framed as a test of inhibition in autism?

There is currently a debate in the literature regarding the extent to which suppressive center-surround phenomena reflect direct GABAergic inhibitory processes. Studies in clinical populations including ASD have assumed (to varying degrees) that differences in suppression reflect disruption of GABAergic inhibition (Tadin et al., 2006; Golomb et al., 2009; Battista et al., 2010; Foss-Feig et al., 2013; Sysoeva et al., 2017; Yazdani et al., 2017; Zhuang et al., 2017). This interpretation is supported by work in mouse models that implicates GABA in surround suppression within visual cortex (Ma et al., 2010; Haider et al., 2010; Adesnik et al., 2012; Nienborg et al., 2013), and to a lesser extent by studies using MR spectroscopy in small samples of human subjects (Yoon et al., 2010; Cook et al., 2016). However, other studies do not support a direct role for GABA in mediating surround suppression (Ozeki et al., 2004; Ozeki et al., 2009; Shushruth et al., 2012; Sato et al., 2016; Liu et al., 2018).

Because increased E/I balance, generally hypothesized to come from decreased inhibition, is a major theory for the neural basis of ASD (Rubenstein, 2003; Yizhar, 2011; Foss-Feig, 2017), we sought to examine the role of GABA in spatial suppression in this disorder using MRS and visual psychophysics in the same subjects. If we had found both weaker spatial suppression and reduced GABA levels in ASD, this

would have supported a role for abnormal GABAergic inhibition in mediating the weaker surround suppression phenomenon in this disorder. Instead, we found no difference in GABA levels between groups, consistent with weaker suppression, but no difference in inhibition in ASD. We have revised the Introduction and Discussion sections to clarify our motivation for examining surround suppression in ASD in the context of inhibition.

2. The computational modeling results, on the other hand, take an unbiased approach to describing the robust behavioral and fMRI evidence presented for weaker spatial suppression in autism. These results follow on from a recent application of the divisive normalization model to other data presenting weaker spatial suppression results in autism (Rosenberg et al., 2015). But Rosenberg et al. reach a different conclusion than the authors present here. The differences between these two papers are not well laid out. The authors should present this previous work upfront, describe the parameters of the divisive normalization model used in Rosenberg et al., 2015 vs in the present work, and justify why the current conclusions presented here differ from those found in Rosenberg et al., 2015.

We now provide a description of the work of Rosenberg and colleagues in the Introduction. We have also revised the Discussion and Supplemental Modeling sections to clarify the differences between our model and other previous models, and to explain why our conclusions differ from previous studies. In brief, we find that narrower top-down gain modulation is sufficient to describe not only our findings of weaker spatial suppression in ASD (Figure 6G-I), but can also be extended to describe the findings of Foss-Feig (2013, as modeled by Rosenberg, 2015; Supplemental Figure 3A-C) and Schauder (2017; Supplemental Figure 3D-F), while their models in our hands are not sufficient to explain our data (Figure 6A-F).

3. For example, Rosenberg et al., 2015 found that a 75% reduction of the suppressive field gain term (relative to controls) describes weaker spatial suppression in autism at high contrasts, but not changes to excitatory gain. Here, the authors chose to test a 25% reduction of the suppressive field gain term relative to controls and reach a different conclusion. Why?

There may be some confusion regarding our terminology, or else our understanding of the work of Rosenberg et al. (2015) does not match the reviewer's. Rosenberg et al. (2015) report in their SI Appendix (pg. 2): "For the simulations presented in Figs. 3, S1, S4, and S5 as well as for the "typically developing control model" in Figs. 4, 5, S2, and S3, the values of these parameters were $v = 1$ and $c = 1 \times 10^{-4}$. In order to increase the E/I ratio, hypothetically simulating autism, a lower value of $c = 7.5 \times 10^{-5}$ was used with $v = 1$ for the "autism model" in Figs. 4, 5, S2, and S3." Thus, our understanding is that for their autism model, the divisive normalization term c was 25% weaker for ASD vs. control models (i.e., 75% as strong), as in our implementation of their model. We have attempted to clarify this in the text.

4. It is difficult to understand the potential impact of the many degrees of freedom in this model on the presented results, especially when authors only present results after tweaking specific parameters to a specific level. To more fully understand the specificity of the presented results to specific levels and parameters, please present a full simulation landscape plotting the size of the suppression index (big – med) as a function of +/- alterations in suppressive gain vs. excitatory spatial filters and suppressive gain vs. top-down gain field size.

We have added a Supplemental Figure (#4) showing the difference in duration thresholds (across stimulus sizes) and size indices predicted by 'control' vs. 'ASD' model variants for weaker spatial suppression, larger excitatory spatial filters, and narrower top-down gain. We also include the observed difference in thresholds between ASD and NT subjects from our psychophysical study, for comparison.

5. Given that neither GABA nor Glx levels predict task performance in either group (controls or autists), it is difficult for the authors to present their MRS data as strong evidence as to the neural basis of their behavioral effects on the tested task. There are many reasons for a lack of correlation between MRS measurements and task performance – ranging from the true dependence of this task on inhibition (as raised above) to the sensitivity of MRS measurements to detect such a dependence in the context of this task and scan protocol. Given that no relationship was observed in this study between MRS measurements and task performance in controls, this aspect of the paper is inconclusive.

Interpreting a null result can be challenging, as the reviewer suggests. Given that we had an a priori hypothesis that weaker suppression in ASD would be linked to reduced GABA levels in visual cortex (which was not confirmed by our data), we may say that our results do not support this hypothesis, and that our data do not indicate any difference in GABA in visual cortex between groups as measured by MRS. We have revised our language in the Results and Discussion to more carefully deal with this issue. Please also see our response to Reviewer 3's first point.

6. The authors have strong, compelling behavioral effects suggesting that spatial suppression is altered in autism at high contrasts. However, this paradigm has previously been explored in autism in previous studies with mixed results (Schauder et al., 2017, Foss-Fieg et al., 2013, etc.). How did these previous results motivate the current work, and how should we interpret the conclusions of these papers? These issues should be discussed in the introduction of the paper, as the previous literature on this paradigm in autism is important context for the reader in interpreting the current results.

We now include a discussion of these previous studies and their role in motivating the current work in the Introduction.

7. The strong behavioral evidence for weaker spatial suppression in autism is reflected in reduced spatial suppression in the BOLD Signal in hMT+. This suggests a brain-behavior link. Does the degree of spatial suppression in behavior predict that in fMRI?

We do not observe a correlation between psychophysical suppression indices and fMRI suppression metrics across individuals ($r_{47} = 0.02$, $p = 0.9$ for all subjects combined, the same is true for each group separately). It is not clear why such a relationship was not found. Possibilities include differences in attention (i.e., gratings were attended during psychophysics, while attention was directed toward the fixation task during fMRI), the involvement of additional brain areas beyond hMT+ (e.g., V1, higher-level regions), the fact that fMRI and psychophysical data were collected in separate experimental sessions, and/or slight stimulus differences (e.g., above-threshold stimulus duration during fMRI). We now include this in the Results section, with the possible explanations for the null result listed above.

8. A large amount of data that was lost in defining foveal hMT+ based on the intersection of a motion localizer and a center-surround checkerboard (10 control participants). Is this because the phase-reversing checkerboards do not optimally drive hMT+? Regardless, given that this definition was apparently not optimal for defining foveal hMT+, what do the results look like when constraining the analysis to hMT as defined by a traditional motion localizer?

The reviewer suggests that the center-surround checkerboard localizer stimulus may not have optimally driven neural responses in hMT+; this is certainly possible. The original intent of the checkerboard localizer was to identify the retinotopic regions in early visual cortex that respond selectively to the foveal center stimuli (and not to the surrounding ring). In our initial analyses, we found that the average fMRI response for all voxels in hMT+ was (perhaps unsurprisingly) much larger for the large vs. small stimulus blocks; this is simply due to the fact that there is a larger stimulus on the screen. Responses in hMT+ voxels with a peripheral retinotopic bias will be more strongly driven by the larger stimuli. Since this pattern of results does not reflect the center-surround effect of interest, we chose to focus on the response within the foveal sub-ROI within hMT+ (based on the additional center-surround checkerboard localizer).

We now include a note in the Methods clarifying that the flickering checkerboard center-surround localizer may not have been optimal for identifying foveal hMT+, and explaining that future studies may benefit from using localizer stimuli better suited for eliciting strong responses in this region (e.g., center & surround defined by drifting gratings or dots).

9. From a brief study of the references cited in the first paragraph of the introduction, I am under the impression that the authors are dramatically oversimplifying the “excitation/inhibition” literature in autism. The results cited in mice and humans often pertain to specific regions of the brain – this is not necessarily evidence for “widespread reduction in inhibition”. Is the theory that certain regions of the brain are affected by reduced neural inhibition in autism? Certain circuits? Is it always reduced inhibition? Recent reviews of the E/I literature in autism and psychiatric conditions highlight the complexities of this theory, which are not captured in the current introduction. Further, given that the paradigm tested here is not really a test of inhibition (see comments above), the current introduction does not seem to accurately introduce the hypotheses tested in this paper relating to a specific neural computation, divisive normalization, in specific region of the autistic brain, hMT+/V1.

We have revised the Introduction and included additional references to more carefully address the complexities of the increased E/I theories of ASD. In particular, we have adjusted the language describing changes in inhibition across cortex, and note that previous studies have found evidence for reduced inhibition in specific brain areas in ASD (e.g., frontal cortex). While reduced inhibition has received particular attention as a theory for the neural basis of ASD (Rubenstein, 2003; Foss-Feig, 2017), we now note in the Introduction and Results that others have instead pointed to increased excitation (Fatemi, 2008; Brown, 2012), which motivated our desire to look at MRS measures of Glx in addition to GABA in the current study.

We have clarified these issues in the Introduction, in addition to the revisions regarding the motivation for using spatial suppression in the context of studying E/I balance (in response to point #1).

Minor comments:

1. The ANOVA reported in the behavioral results section - are size and contrast both included as within-subject variables?

Yes, both of these factors were included as within-subjects variables in the ANOVA examining motion duration thresholds; this has been clarified in the Methods.

2. Model – Is the spatial extent of inhibition also impacted by altering the amplitude of the inhibitory field in the current model?

For the ‘weaker normalization’ model illustrated in Figure 6A-C (a la Rosenberg), reducing the suppressive gain factor necessarily shrinks the effective size of the suppressive drive in both the spatial and orientation dimensions, as scaling down a 2D Gaussian brings all values closer to zero multiplicatively. This has been clarified in the text. We hope that by publishing our Matlab code alongside this manuscript, the reader will be better able to understand the impact of changing different model parameters.

3. fMRI – How big are the bilateral hMT+ ROIs, and are the two groups matched in size (voxel #)?

ROIs for hMT+ in each hemisphere were defined at the individual subject level by finding the top 20 voxels whose time course was most highly correlated with the predicted time course (block design matrix convolved with a canonical HRF), within significantly activation regions of the lateral occipital lobe ($p < 0.05$, Bonferroni corrected). This procedure is described in the “Data analysis and statistics” section, and we have revised the description to improve the clarity. Therefore, the same number of voxels (40) are included for each subject.

4. fMRI – the suppression effect is measured by taking BOLD response in foveal hMT+ to large – the preceding small stimuli. Results show weaker suppression in autism given this baseline. How do baseline responses (to the preceding small stimuli) compare between the groups?

Because there is no ‘rest’ condition in our fMRI task (i.e., there is always a drifting grating stimulus on the screen), it is not possible to quantify the response to the small stimuli independently from the response to the larger stimuli in this paradigm, one must serve as the baseline for the other. We have clarified this in our description of the task in the Methods section.

In a separate set of experiments in subjects with and without ASD, we have quantified the fMRI response in visual areas to small drifting gratings relative to a no-stimulus baseline (i.e., ‘rest’). The findings from these experiments have been described in a separate manuscript (recently submitted for publication), as they are not directly relevant to the spatial suppression phenomenon which is the focus of the current study.

5. MRS/fMRI - No details are given regarding MRS data quality (field drift, spectral line width, SNR, etc.) or fMRI data quality (in-scanner motion), as compared across groups.

We now provide information about MRS and fMRI data quality comparisons between groups in the “Control analyses” section of the results, with full details in the Supplemental Information.

We did not see significant differences between ASD and NT participants in frequency variability of water throughout the scan, number of TRs rejected for artifacts during frequency correction, or Glx fit residuals in either hMT+ or EVC (Mann-Whitney tests, Z-values < 1.53, uncorrected p-values > 0.126, data not shown). The residual signal for GABA in hMT+ after fitting was higher for ASD (median = 5.2) versus NT participants (median = 4.7, Mann-Whitney test, Z = 2.21, uncorrected p = 0.027), and the spectral width of the creatine signal (FWHM) in EVC was broader for ASD participants (median = 8.7 Hz) versus NTs (8.3 Hz; Mann-Whitney test, Z = 2.48, uncorrected p = 0.013), suggesting somewhat lower data quality in participants with ASD on these metrics. However, neither of these were significant following correction for multiple comparisons (FDR corrected p = 0.24 and p = 0.13, respectively). There were no significant group differences for GABA residuals in EVC or creatine FWHM in hMT+ (Mann-Whitney tests, Z-values < 1.59, uncorrected p-values > 0.112). Thus, we do not find strong evidence to suggest that systematic differences in MRS data quality between groups may have greatly impacted the observed pattern of results.

Head motion during fMRI (mean framewise displacement [FD]) was significantly greater among participants with ASD vs. NTs (ASD mean = 0.15 mm, SD = 0.09, NT mean = 0.09 mm, SD = 0.05; Mann-Whitney test, Z = 2.64, p = 0.008; data not shown). However, because our fMRI results were qualitatively the same (weaker spatial suppression in hMT+ among ASD participants) after excluding data segments and subjects with excessive head motion (FD > 0.9 mm; see Supplemental Control Analyses, Supplemental Figure 5), we do not believe that our fMRI results may be explained by group differences in head motion.

Reviewer #3 (Remarks to the Author):

Review of Nature Comms.19-17709 Weaker Neural Suppression in Autism

The authors present a compelling multimodal study of fMRI and MRS in a well-grounded and theoretically-principled experimental design. They then invoked a computation model to account for observed data.

Consistent with prior studies of cortical GABA estimation in visual cortex (e.g. Gaetz et al.), no differences from typical levels are observed in ASD in this study, despite weaker neural suppression (measured behaviorally and by fMRI). They introduce a computational model and propose that weaker neural suppression could be attributable to differences in top-down processing, which is certainly interesting. A more trivial explanation could, however, lie in either the imprecision of GABA estimation per se, or the lack of specificity to the nature of GABA being estimated (since voxels are rather large compared with cellular compartments), leaving most to interpret such MRS-based GABA estimates (at best) as indices of inhibitory tone. While that could be addressed through slightly more conservative discussion of the interpretation of the findings, perhaps a more compelling experiment would be to attempt analogs in other cortices (if conceivable) where similarly-obtained GABA estimates

have consistently shown deficits in ASD (e.g. sensorimotor, Gaetz et al., Puts et al. or auditory (Rojas et al., Gaetz et al., Port et al.).

We have revised the Discussion to more carefully address the issues raised regarding the interpretation of our GABA MRS results.

In a separate experiment, we have obtained GABA MRS measurements in subjects with ASD and NTs across a range of cortical regions (visual, auditory, somatosensory); our findings are described in a separate manuscript (recently submitted for publication), as they do not directly concern neural suppression in these areas (unlike the current study). We now note in the Discussion of the current manuscript that future studies may benefit from examining neural suppression in a diverse set of brain regions, as suggested by the reviewer.

Minor points –

it is unfortunate that the NT group did not also undergo ADOS observation – how sure are we that the controls exhibit no ASD-tendencies ?

All subjects were assessed by clinicians with extensive experience with ASD, under the supervision of a doctoral-level clinical psychologist who had achieved research reliability in the gold standard tools used to diagnose ASD. In the course of their assessment, clinicians referenced DSM-5 criteria to rule-out ASD based on clinical observation. This has been clarified in the Methods. Indeed, one subject who initially enrolled as a NT individual (and did not endorse an ASD diagnosis) was suspected of having ASD during our clinical evaluation. The subject was later found to meet ASD diagnostic criteria (confirmed using ADOS-2 and ADI-R) and was subsequently included in the ASD group.

We also collected the Social Responsiveness Scale, 2nd Ed. (SRS-2) for all participants, including NTs. SRS-2 total scores were below clinical cutoff scores for all of our NT participants. SRS-2 scores are now reported for both groups in Table 1.

The proportion of males and females is similar in the groups, which is commendable. Is there sufficient sample to estimate a sex effect in the neural suppression or GABA levels (or both) ?

In our “Control analyses” sections (Results and Supplemental Information) we used linear mixed effects model analyses to show that demographic factors (including sex) did not account for the observed differences in psychophysical and fMRI suppression between ASD and NT subjects.

We have added more details to this section regarding the main effects of sex. Briefly, we found a significant main effect of sex on psychophysical suppression indices ($t_{497} = -2.17$, $p = 0.030$), but no effect of sex on fMRI suppression in hMT+ ($t_{93} = 0.739$, $p = 0.5$).

We now also include additional linear mixed effect model results examining the effect of demographic factors on GABA and Glx values from MRS. Including demographic factors did not reveal any significant

main effects of diagnostic group, and no significant main effects of sex were observed for GABA or Glx in either hMT+ or EVC.

NVIQ (although not different between groups) seem rather high to be broadly representative.

Please see our response to Reviewer 1's point #1.

It seems the GABA single voxel was placed according to fMRI results – necessitating the EPI run to precede the MRS – did the authors encounter any (very typical) issues with eddy current heating (and subsequently cooling) leading to field drift across the (11-minute) MRS acquisition? Were the sequence of voxel placements randomized across subjects? They argue that the fMRI preceded the anatomic scan too (was that sufficient to prevent field drift? this is easily ascertained since 3 MRS acquisitions were performed – was the center frequency constant at the beginning and end of each acquisition)?

The reviewer is correct that we chose to run the T1 anatomy scan after the fMRI functional localizer, in order to allow the gradients to cool as much as possible prior to MRS acquisition. We also chose to use a relatively low-power fMRI sequence for our localizer (3 s TR, 3 x 3 x 5 mm resolution, 14 slices) for the same purpose. We used a fixed acquisition order for MRS scans (left hMT+ first, EVC in the middle, right hMT+ last), to ensure that any effects of gradient heating would be equivalent within a given voxel across all subjects and groups. This detail is now clarified in the Methods. We did not observe any difference in water frequency drift (SD in Hz across the scan) when comparing the first (left hMT+; mean = 1.09 Hz) and last (right hMT+; mean = 0.99 Hz) MRS scans ($t_{61} = 1.17$, $p = 0.25$), thus we do not believe that there were systematic differences in data quality between these two runs due to gradient heating. This is now also clarified in the Methods.

Did GABA levels drift during the course of the fMRI stimulation paradigm? While I am sure GABA and fMRI were not interleaved acquisitions (although this might be an interesting direction), was GABA estimation performed both before and after the visual paradigm (at least for the a priori selected EVC voxel placement)?

MRS and fMRI data were acquired in different scanning sessions, in many cases on different days. We did not acquire MRS data both before and after fMRI in any sessions, so we are not able to address this question directly, though we agree that it is an interesting future direction (provided one adequately deals with issues of gradient heating, as discussed above).

Similarly it seems strange to allow the participants to watch different movies during GABA estimation in visual cortex – presumably GABA levels are approximately constant and insensitive to the nature of visual stimulation, but I would have thought no stimulation (or at least a constant level of luminance/contrast/dynamics would have been preferred).

We agree that having no stimulus could help avoid possible confounds of metabolite levels fluctuating during a scan. However, participants from clinical populations such as ASD may be more likely to abort a scan early when they are uncomfortable, and we have found that watching a movie can help subjects better tolerate long (>1 hour) scanning sessions. Thus, we decided to allow subjects to watch movies during the MRS in order to maximize subject compliance, and to minimize boredom and fatigue, which can both lead to increased head motion. This is now clarified in the Methods.

What is the nature of the small peak at ~3.2ppm? It is excluded from analysis. Can the authors be sure it is not attributable to GABA? The side-band of a GABA triplet might be expected to be about .2ppm apart I suppose. Or is it just an irrelevant metabolite that should indeed be discarded ? Does it contaminate the GABA estimates (i.e. should it be fitted out separately) ? On that point, it would probably be helpful to see the individual fits as a stacked plot. GANNET 2.0 fitting is extremely sensitive to any baseline fluctuation around 3-3.5ppm (since it fits a “Gaussian plus a line”).

The reviewer makes an astute observation; the small unfit peak at 3.2 ppm is thought to reflect a small co-edited Choline signal that may be left over following the subtraction of edit ‘on’ and edit ‘off’ MEGA-PRESS scans (Evans et al., 2012, “Subtraction artifacts and frequency (Mis-)alignment in J-difference GABA editing.” Journal of Magnetic Resonance Imaging). This peak is outside the known range of GABA signals, and as such is not modeled by GANNET 2.0. Although we do not expect a dramatic effect on GABA quantification, we hope to explore the idea of explicitly fitting this peak in our future work.

To the reviewer’s second point, we now show stacked plots of the spectra from all subjects, in addition to the zoomed-in mean + SD plots in Supplemental Figure 5.

Reviewers' Comments:

Reviewer #1:

Remarks to the Author:

I'm grateful to the authors for considering my suggestions, and for their constructive and helpful responses. These have addressed my concerns and improved the manuscript. I have no further concerns or suggestions.

Reviewer #2:

Remarks to the Author:

The authors have made many improvements to the manuscript in this revision. Unfortunately, a number of concerns remain regarding the novelty of these findings and the interpretation of the results. As it currently stands, I do not think the paper is suitable for publication in Nature Communications.

Major concerns:

1. To my eye, the key link the authors wish to make in this study simply didn't work out. The initial hypothesis was that GABA or Glx levels in controls would predict performance on the spatial suppression task. They didn't: the authors show that these signals are not task-relevant. (Neither gaba nor glx predict performance in controls). So why do the authors continue to argue that GABA or Glx concentrations are relevant to understanding the performance of the autistic group on this same task? The initial hypothesis was well-motivated by computational models of spatial suppression. But given the results, the argument as it stands -- that "weaker neural suppression in ASD may be attributable to differences in top-down processing, but not to differences in GABA levels" -- is unsound. GABA levels (at least as measured using MRS in humans) are undermined as relevant to understanding spatial suppression by the presented control data. This is not necessarily surprising for, as reviewer 3 notes, the origin of the MRS GABA signal is not clear.
2. The divisive normalization model used in this study has been previously employed by Rosenberg et al., 2015 to model the same visual phenomenon (weaker spatial suppression in autism). Yet, the modeling results produced by the Schallmo et al. and Rosenberg et al. are different from one another. Why? The authors do not explain how two groups using the same model to describe the similar behavioral findings can reach opposite conclusions. Moreover, the fact that this analysis is a direct attempt to replicate Rosenberg et al., which arguably fails, is not clearly laid out in the manuscript.
3. I thank the authors for providing Supplemental Figure 4, which shows the relative effects of altering (a) the suppressive gain term, (b) excitatory sf size, or (c) top-down width on group differences in spatial suppression. This figure is difficult to understand based on the legend, but to my eye, similar results are obtained by altering either a or c, as consistent by Rosenberg et al. Again, I am not convinced that a strong conclusion regarding the selectivity of this group difference to top-down modulation (over suppressive gain as reported by Rosenberg et al.) can be drawn from the modeling results.
4. The strength of the conclusions stated in this paper are surprising considering that the main behavioral finding of this paper (weaker spatial suppression in autism) has been not-replicated twice (Schauder et al., 2017, Sysoeva et al., 2017). Sysoeva et al. is not even referenced in the current MS, and the mixed results in the literature is not transparent. A reader of this manuscript would not be able to easily take away from this paper that the primary behavioral difference reported in this paper has been non-replicated twice in the literature. I suggest that a future revision of this paper emphasize the replication of previous behavioral findings (Foss Fiegg et al., 2017), despite other failed replication attempts (Schauder et al., 2017, Sysoeva et al., 2017), as well as the novel contribution of

a fMRI component to this growing body of literature, which provides a more conclusive piece of evidence than previously available in the behavioral literature alone.

Reviewer #3:

None

We would like to thank both of the reviewers for their comments. We understand and appreciate the feedback from Reviewer 2, and we have sought to fully address every point of concern that was raised through a major revision to our manuscript. The revised text has been indicated in red.

We have prepared the following point-by-point response:

Reviewer #1 (Remarks to the Author):

I'm grateful to the authors for considering my suggestions, and for their constructive and helpful responses. These have addressed my concerns and improved the manuscript. I have no further concerns or suggestions.

Reviewer #2 (Remarks to the Author):

The authors have made many improvements to the manuscript in this revision. Unfortunately, a number of concerns remain regarding the novelty of these findings and the interpretation of the results. As it currently stands, I do not think the paper is suitable for publication in Nature Communications.

Major concerns:

1. To my eye, the key link the authors wish to make in this study simply didn't work out. The initial hypothesis was that GABA or Glx levels in controls would predict performance on the spatial suppression task. They didn't: the authors show that these signals are not task-relevant. (Neither gaba nor glx predict performance in controls). So why do the authors continue to argue that GABA or Glx concentrations are relevant to understanding the performance of the autistic group on this same task? The initial hypothesis was well-motivated by computational models of spatial suppression. But given the results, the argument as it stands -- that "weaker neural suppression in ASD may be attributable to differences in top-down processing, but not to differences in GABA levels" -- is unsound. GABA levels (at least as measured using MRS in humans) are undermined as relevant to understanding spatial suppression by the presented control data. This is not necessarily surprising for, as reviewer 3 notes, the origin of the MRS GABA signal is not clear.

The reviewer is correct to assume that the original hypothesis that motivated these experiments focused on the role of GABA levels (and secondarily, Glx) in spatial suppression. We agree that it is not appropriate to have GABA, glutamate, and E/I balance as major points of focus for the manuscript, given that we found no significant group differences, nor any association between the metabolite measures from MR spectroscopy and either task performance or fMRI responses. We have substantially revised the manuscript in the following manner to address this issue:

- a. In order to de-emphasize this aspect of the study, we have greatly condensed the presentation of the MR spectroscopy findings in the Results, and moved most of this material (including the figure) to the Supplemental Information. We have also substantially revised the language and interpretation of these findings throughout the manuscript, to reflect that no strong conclusions may be drawn from the non-significant correlations between metabolite levels and suppression metrics.

- b. We have removed discussion of E/I balance from the Introduction and Discussion, and re-focused the text on neural regulatory mechanisms (i.e., center-surround processing) and their disruption in ASD.

2. The divisive normalization model used in this study has been previously employed by Rosenberg et al., 2015 to model the same visual phenomenon (weaker spatial suppression in autism). Yet, the modeling results produced by the Schallmo et al. and Rosenberg et al. are different from one another. Why? The authors do not explain how two groups using the same model to describe the similar behavioral findings can reach opposite conclusions. Moreover, the fact that this analysis is a direct attempt to replicate Rosenberg et al., which arguably fails, is not clearly laid out in the manuscript.

We have substantially revised the Introduction, Results, and Discussion to fully clarify the distinctions between our results and those of Rosenberg and colleagues (2015), and our motivation for presenting an alternative model within the divisive normalization framework. In particular, we have taken care to delineate:

- a. The motivation for our modeling work was not to replicate Rosenberg's work as such, but to find a computational account that could satisfactorily describe our behavioral results as well as those of previous studies (i.e., Foss-Feig, 2013; Schauder, 2017).
- b. We found that weaker spatial suppression in ASD (as we observed both behaviorally and with fMRI in hMT+) could not be accounted for by the weaker normalization model of Rosenberg and colleagues (2015). Although weaker normalization yields lower predicted motion duration thresholds (i.e., better performance), reducing the strength of normalization has little or no effect on spatial suppression (i.e., the difference in thresholds for smaller vs. larger stimuli; see Figure 5C [was previously Figure 6C] and Supplemental Figure 5D [previously Supplemental Figure 4C]). To clarify this point, we have added red arrows to Figure 5, indicating places where the predicted thresholds or size indices from the Rosenberg and Schauder models fail to match the behavioral results we observed (Figure 2A-C).
- c. Whereas previous models from Rosenberg (2015) and Schauder (2017) point to *low-level* differences in neural processing (weaker normalization and larger excitatory spatial filters, respectively), our model characterizes differences in motion duration thresholds in ASD in terms of a difference in *higher-level*, top-down modulation.
- d. In contrast to Rosenberg (2015), our model is able to simulate conditions under which thresholds are *higher* in the ASD group (as reported by Schauder et al. (2017) and Sysoeva et al. (2017); Supplemental Figure 4E).
- e. We have made the code for our computational modeling work publicly available on GitHub (github.com/mpschallmo/WeakerNeuralSuppressionAutism). This link is now provided in the Code Availability section of the Methods. This code includes both our novel narrower top-down modulation model, as well as our implementations of the models proposed by Rosenberg (2015) and Schauder (2017). We hope that by making our code more easily accessible, we may help to obviate any difficulty in understanding the details of the different model variants we have examined.

3. I thank the authors for providing Supplemental Figure 4, which shows the relative effects of altering (a) the suppressive gain term, (b) excitatory sf size, or (c) top-down width on group differences in spatial suppression. This figure is difficult to understand based on the legend, but to my eye, similar results are obtained by altering either a or c, as consistent by Rosenberg et al. Again, I am not convinced that a strong conclusion regarding the selectivity of this group difference to top-down modulation (over suppressive gain as reported by Rosenberg et al.) can be drawn from the modeling results.

We understand the difficulty in deciphering Supplemental Figure 4 (now Supplemental Figure 5 in the revised manuscript), and acknowledge that the information it contains is densely encoded. We have taken the following steps to clarify this figure, and to highlight where panels B-D (Rosenberg, previously A-C) and H-J (current model, previously G-I) differ:

- a. We now provide a graphical depiction of the method for generating the data surfaces (now panel A), showing that the model was run after changing a given parameter value, and then the predicted thresholds (at all stimulus sizes) for this model variant were subtracted from the thresholds predicted by the "base" model (which was designed to simulate thresholds from the NT group). In this way, we attempted to model a possible group difference (i.e., NT - ASD thresholds). The differences in thresholds for this model variant were then entered as a row in the model surface (heatmap), with different parameter values shown along the y-axis of the surface.
- b. We now include green arrows to indicate places where particular models do match our observed behavioral results (in addition to the red arrows indicating a poor match).
- c. We have provided additional description and interpretation of these findings in the Results and Supplemental Information. Briefly, we have clarified how this figure illustrates that the models of Rosenberg (2015) and Schauder (2017) fail in different ways to account for the behavioral findings we observed in the current study, while our proposed model (i.e., narrower top-down modulation) provides a better account for our psychophysical results. With regards to the Rosenberg model, weaker spatial suppression is *not* observed when reducing normalization strength (panel D), whereas our narrower top-down modulation model does predict weaker spatial suppression (panel J).

4. The strength of the conclusions stated in this paper are surprising considering that the main behavioral finding of this paper (weaker spatial suppression in autism) has been not-replicated twice (Schauder et al., 2017, Sysoeva et al., 2017). Sysoeva et al. is not even referenced in the current MS, and the mixed results in the literature is not transparent. A reader of this manuscript would not be able to easily take away from this paper that the primary behavioral difference reported in this paper has been non-replicated twice in the literature. I suggest that a future revision of this paper emphasize the replication of previous behavioral findings (Foss Fiegg et al., 2017), despite other failed replication attempts (Schauder et al., 2017, Sysoeva et al., 2017), as well as the novel contribution of a fMRI component to this growing body of literature, which provides a more conclusive piece of evidence than previously available in the behavioral literature alone.

We appreciate and agree with the Reviewer's suggestion to more carefully address the disparate findings in the literature, and we thank them for noting our omission of the Sysoeva (2017) paper. As part of our revisions of the Introduction and the Discussion we now provide a more thorough explanation of previous disparate behavioral findings, and describe how our results fit into this literature, in addition to tempering the strength of our conclusions. We also highlight the novelty and importance of our fMRI results showing weaker suppression in hMT+ in ASD. Finally, we have more carefully presented our modeling results as an attempt to account for the various behavioral findings (ours and others) which do not agree, under a single computational framework that may explain both superior and impaired motion discrimination in ASD.

Finally, we have clarified that our data from this study are available from the NIMH Data Archive (nda.nih.gov/edit_collection.html?id=2266). This link is now provided in the Data Availability section of the Methods.

Reviewers' Comments:

Reviewer #2:

Remarks to the Author:

I am grateful to the authors for their constructive responses to my concerns and the substantial revisions of their manuscript. I feel that all of the points I raised have been thoroughly addressed and have no further concerns.